# InertialAR: Autoregressive 3D Molecule Generation with Inertial Frames

**Haorui Li** [1]  **Weitao Du** [2]  **Yuqiang Li** [3]  **Hongyu Guo** [4]  **Shengchao Liu** [1]

## Abstract

Transformer-based autoregressive models have emerged as a unifying paradigm across modalities such as text and images, but their extension to 3D molecule generation remains underexplored. The gap stems from two fundamental challenges: (1) how to tokenize molecules into a canonical 1D sequence of tokens that is invariant to both SE(3) transformations and atom index permutations, and (2) how to design an architecture capable of modeling hybrid atom-based tokens that couple discrete atom types with continuous 3D coordinates. To address these challenges, we introduce InertialAR. It first performs generation-oriented canonical tokenization by aligning each molecule to a canonical inertial frame and reordering atoms, thereby converting arbitrary 3D structures into a unique, SE(3)- and permutation-invariant sequence of tokens for autoregressive generation. Built upon this canonical tokenization, we propose geometric positional encoding (GeoPE), which endows Transformer attention with 3D geometric awareness. Finally, InertialAR utilizes a hierarchical autoregressive paradigm to decode the next atom, consecutively predicting the atom type and 3D coordinates via Diffusion Loss. Experimentally, InertialAR achieves state-of-the-art performance on 8 of the 10 evaluation metrics for unconditional generation across QM9, GEOM-Drugs, and B3LYP. Moreover, it significantly outperforms baselines in controllable generation for targeted chemical functionality, attaining state-of-the-art results across all 5 metrics. Code is available at github.com/HaoruiLi46/InertialAR.

## 1. Introduction

Autoregressive (AR) models have achieved substantial progress in artificial intelligence (AI) in recent years. In natural language processing, their strong sequence modeling capability and scalability have established them as the de facto architecture for foundation models (Brown et al., 2020; Touvron et al., 2023; Achiam et al., 2023). Moreover, they have shown competitive performance on par with diffusion models in image generation, suggesting their viability as a unified sequence modeling paradigm (Sun et al., 2024; Tian et al., 2024). Inspired by their success across these diverse modalities, we seek to investigate whether AR models can serve as an effective generative model paradigm for 3D molecule generation.

While diffusion models have achieved impressive results in 3D molecule generation, they are often limited by computationally intensive iterative sampling and a lack of flexibility for variable-length generation (Hoogeboom et al., 2022; Xu et al., 2023; Vignac et al., 2023). In contrast, AR models offer a compelling alternative: by casting 3D molecule generation as a sequence prediction problem, they enable highly efficient and flexible generation of variable-sized molecules.

However, adapting AR models for 3D molecule generation poses unique challenges at both data and model levels. On the data side, the key difficulty centers on tokenizing 3D molecules into 1D sequences of tokens compatible with Transformer-like models. An ideal tokenization must satisfy two criteria: (1) SE(3)-invariance, *i.e.*, invariant tokenization under rotations and translations, and (2) permutation invariance of the atom indexing to establish a canonical sequence order for each molecule. On the model side, unlike conventional AR models that merely predict the next discrete token at each step, the AR model for 3D molecule generation requires jointly predicting a discrete atom type and its continuous 3D coordinates, due to the dual chemical and geometric information encoded in each atom.

**Our Contributions.** To address these challenges, we propose InertialAR, a novel autoregressive model for 3D molecule generation (Figure 1). First, we introduce generation-oriented canonical tokenization: aligning molecules to a canonical inertial frame ensures SE(3) invariance, while deterministic atom reordering guarantees permutation invariance. Together, they convert arbitrary 3D

[1] The Chinese University of Hong Kong [2] Alibaba DAMO Academy [3] Shanghai Artificial Intelligence Laboratory [4] University of Ottawa. Correspondence to: Shengchao Liu <scliu@cse.cuhk.edu.hk>.

*Proceedings of the 43rd International Conference on Machine Learning*, Seoul, South Korea. PMLR 306, 2026. Copyright 2026 by the author(s).

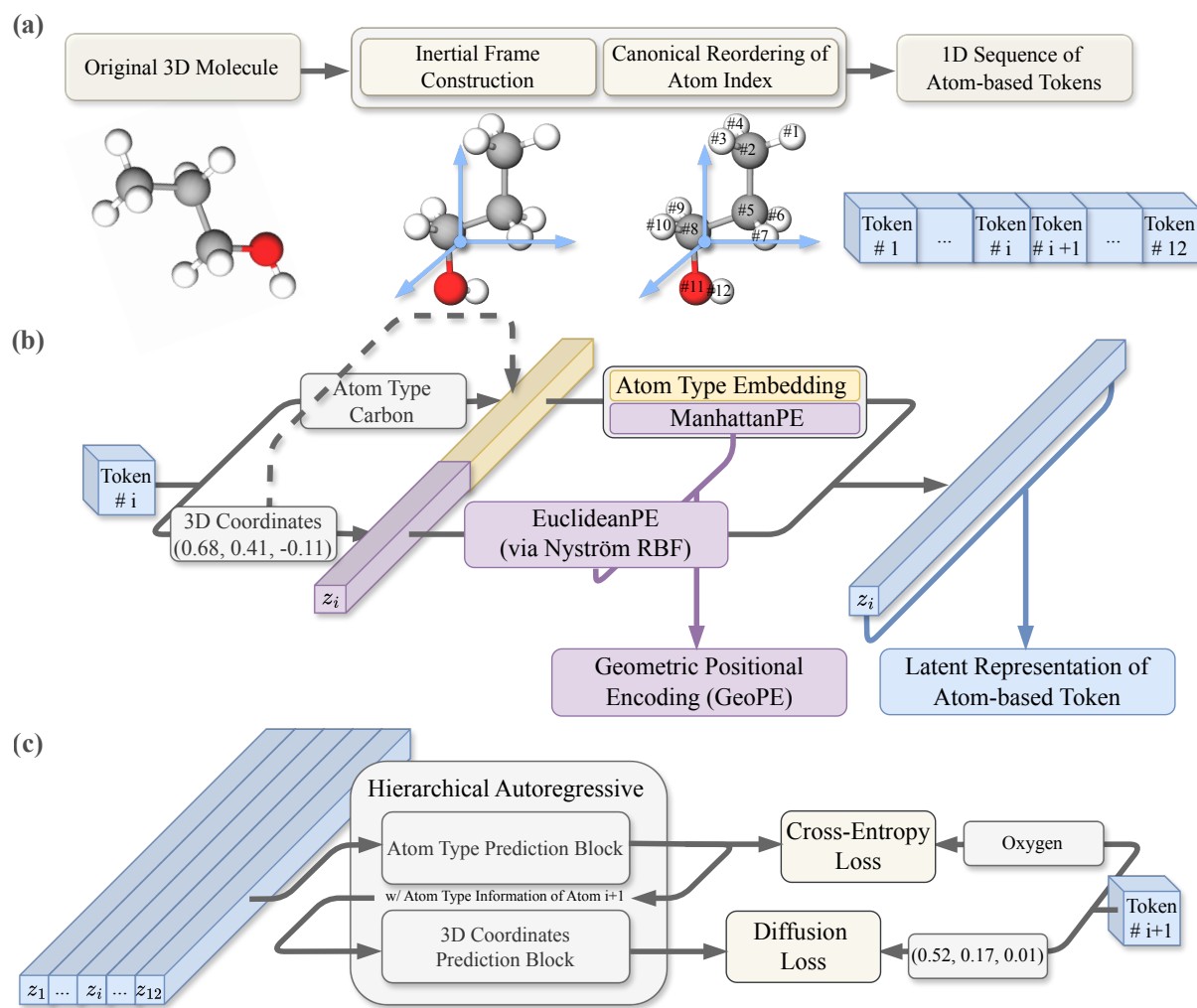

*Figure 1.* Overview of InertialAR: (a) canonical tokenization, (b) geometric positional encoding, and (c) hierarchical AR paradigm.

structures into a unique sequence of atom-level tokens for autoregressive generation. Although inertial frames have appeared in molecular modeling, to the best of our knowledge, this is the first work that turns inertial-frame canonicalization into a generation-ready tokenization in service of autoregressive 3D generation. Second, built upon this canonical tokenization, we propose geometric positional encoding (GeoPE), which injects both Manhattan-style relative positional awareness and pairwise Euclidean distance information between atoms into the attention mechanism, making it geometry-aware. Finally, to handle the hybrid discrete-continuous nature of atom-based tokens, InertialAR adopts a hierarchical AR paradigm: predicting atom types first via cross-entropy, then 3D coordinates via Diffusion Loss.

To evaluate the effectiveness of InertialAR, we conduct comprehensive experiments on both unconditional and controllable generation. For unconditional generation, InertialAR achieves state-of-the-art results on 4 of 6 key metrics on QM9 and GEOM-Drugs. To further assess its scalability and robustness, we evaluate on the more challenging large-scale B3LYP dataset, where InertialAR attains state-of-the-art performance across all 4 metrics, clearly surpassing other prominent diffusion and AR models. Furthermore, on the more demanding task of class-conditional generation, InertialAR combined with classifier-free guidance establishes state-of-the-art results on all 5 evaluation metrics, enabling targeted generation and editing of molecules with desired chemical functionality.

**Related Work** We briefly review the most related works here and include a more detailed overview in Appendix A. The central requirement for 3D molecule generation is respecting SE(3) symmetry. Existing methods can be grouped into four paradigms: (i) SE(3)-equivariant architectures (Thomas et al., 2018; Liao & Smidt, 2023; Schütt et al., 2021; Satorras et al., 2022b), (ii) invariant-feature

modeling (Schütt et al., 2017; Gasteiger et al., 2022), (iii) data augmentation (Flam-Shepherd & Aspuru-Guzik, 2023; Abramson et al., 2024), and (iv) input canonicalization (Antunes et al., 2024; Li et al., 2024b; Fu et al., 2024). Another key challenge for autoregressive 3D generation is tokenization. While recent studies have investigated text sequence-based tokenization (Li et al., 2024b; Yan et al., 2024; Flam-Shepherd & Aspuru-Guzik, 2023), some parallel works are concurrently exploring voxel-based approaches (Faltings et al., 2025; Lu et al., 2025b). However, they both rely on spatial discretization, which discards fine-grained geometry and fails to preserve atom-level granularity.

## 2. Preliminaries

**3D Molecule Generation.** The goal of 3D molecule generation is to directly construct physically plausible 3D molecular conformations. Formally, a 3D molecule with $n$ atoms can be represented as a point cloud $\mathcal{M} = (t, C)$. The vector $t = [t_1, \cdots, t_n] \in \mathbb{Z}^n$ encodes the atom types (atomic numbers), and the coordinate matrix $C = [c_1, \cdots, c_n] \in \mathbb{R}^{3 \times n}$ specifies the 3D position of each atom, with $c_i \in \mathbb{R}^3$.

**Autoregressive Models and Tokenization of 3D Molecules.** Autoregressive models solve sequence modeling by framing it as a "next-token prediction" problem. This approach, an application of the chain rule of probability, factorizes the joint distribution of a sequence $x = (x_1, \ldots, x_n)$ into a product of conditional probabilities:

$$p(x) = p(x_1, \ldots, x_n) = \prod_{i=1}^{n} p(x_i | x_1, \ldots, x_{i-1}). \quad (1)$$

The model's core task is thus to learn the conditional distribution $p(x_i | x_{<i})$ for each step, which is typically parameterized by a powerful neural network such as the Transformer (Vaswani, 2017). The primary challenge in applying AR models to 3D molecular generation lies in the effective tokenization of a 3D molecular structure into a 1D sequence of tokens suitable for Transformer architectures.

**Class-conditional Generation and Classifier-free Guidance.** Class-conditional generation produces samples conditioned on a class label $c$ (Esser et al., 2021; Peebles & Xie, 2023). Classifier-free guidance (CFG), originally proposed by Ho & Salimans (2022), enhances both sample quality and conditional alignment. It trains a single model on both the conditional distribution $p(x|c)$ and the unconditional distribution $p(x)$ by randomly dropping labels during training. Then during inference, conditional generation is steered by combining the two predictions:

$$p_g = p_u + s(p_c - p_u), \quad (2)$$

where $p_c$ and $p_u$ denote the conditional and unconditional

predictions, respectively, and $s$ is the guidance scale controlling the trade-off between class fidelity and sample diversity.

## 3. InertialAR

The Inertial Autoregressive Model (InertialAR) casts 3D molecule generation as an AR process, where a molecule is sequentially built by predicting "the next atom-based token" at each step. To achieve this, a 3D molecule $\mathcal{M}$ is tokenized into an ordered 1D sequence of $n$ atom-based tokens, $\mathcal{M} = (a_1, \ldots, a_n)$, where each atom-based token $a_i = (t_i, c_i)$ contains a discrete atom type $t_i$ and continuous 3D coordinates $c_i = (x_i, y_i, z_i)$. Thus, the corresponding probability factorizes as:

$$p(\mathcal{M}) = \prod_{i=1}^{n} p(a_i | a_{<i}) = \prod_{i=1}^{n} p((t_i, c_i)|a_{<i}). \quad (3)$$

### 3.1. Generation-oriented Canonical Tokenization

The factorization in Equation (3) makes AR models inherently sensitive to the token order. Therefore, a robust tokenization must be invariant to two fundamental symmetries: the continuous SE(3) symmetry of the molecular geometry under rotations and translations, and the discrete permutation symmetry of the atom indexing (which can yield up to $n!$ permutations for $n$ atoms). Such a canonical tokenization ensures that each molecule maps to a unique token sequence, eliminating ambiguity and enabling effective learning.

More concretely, we introduce generation-oriented canonical tokenization via a two-step procedure, as shown in Figure 1(a). First, to address SE(3) symmetry, we align the molecular system to its canonical inertial frame, resulting in an invariant canonical pose. Second, to address index permutation symmetry, the atoms are deterministically reordered according to a predefined rule. More details are explained below.

**Step 1: Canonical Inertial Frame Construction**. First, we employ the following steps to derive the reference frames that construct the rotation matrix from $N$ atomic positions $c$: (1) Calculate the center of mass: $\bar{c} = \frac{1}{N} \sum_i c_i$. (2) Adjust position relative to the center $c_i = c_i - \bar{c}$. (3) Compute the inertia tensor $\hat{I} = \sum_i (\|c_i\|^2 I - c_i c_i^T)$, where $I$ is the unit diagonal matrix.

**How to define the orderings of canonical inertial frame axes?** We diagonalize the inertia tensor to obtain eigenvalues $\lambda_1 \le \lambda_2 \le \lambda_3$ with corresponding eigenvectors $e_1, e_2, e_3$, which are assigned to the $x$-, $y$-, and $z$-axes.

**How to define the directions of canonical inertial frame axes?** Let $E = [e_1, e_2, e_3]$ denote the orthonormal matrix formed by these eigenvectors. The orthonormal matrix $E$ serves as the coordinate basis. Meanwhile, there are eight

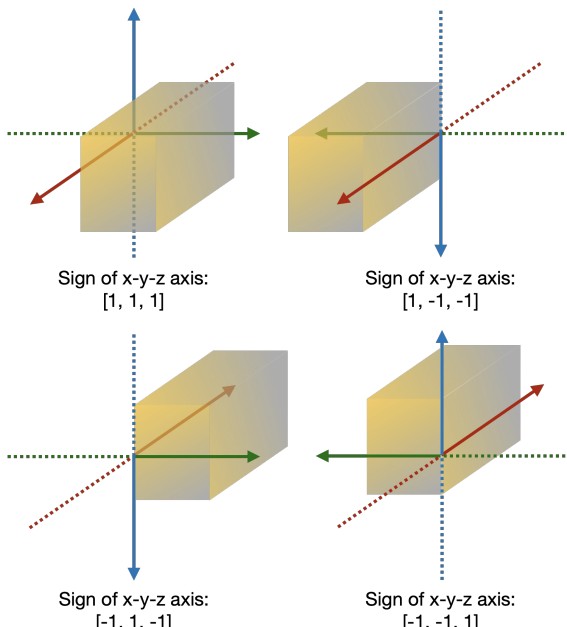

*Figure 2.* Illustration of introducing a fourth node as the anchor node. We define the sign of the x-y-z axis to make sure that $x_4$ is in the first quadrant, and there are four cases in the four subfigures.

possible sign combinations for the $x$-, $y$-, and $z$-axes, given by $\{\pm1, \pm1, \pm1\}$, respectively. First, we enforce a right-handed coordinate system, *i.e.*, the determinant of $E$ to be $+1$, not $-1$. This still leaves four possible sign assignments.

**Sign disambiguation under generic conditions.** Once the principal-axis ordering is fixed, a fourth point with nonzero $x$- and $y$-coordinates in the principal-axis basis provides a sufficient condition for resolving the residual sign ambiguity. In particular, under distinct principal moments and deterministic anchor selection, requiring the anchor to lie in the first quadrant uniquely determines the sign assignment among the four right-handed candidates. Appendix D provides a formal statement and proof under these assumptions.

In practice, we therefore include a fourth node to resolve the directions of the three axes. To achieve this, we consider a fourth node $x_4$ that is not on the y-z plane or x-z plane and has the largest distance to the origin. We then define the *requirement* that x-$x_4$-z and $x_4$-y-z are also right-handed; in other words, this requirement is essentially saying that $x_4$ should be in the first quadrant of the x-y plane. For implementation, $x_4$ is a 3D point whose projection onto the x–y plane falls into one of the four quadrants: the first, second, third, or fourth quadrant, depending on the signs of its x and y coordinates. Each of them defines the signs (or directions) of the canonical inertial frame axes (Figure 2).

**Empirical Robustness of Inertial Frames.** Inertial frame canonicalization has two known theoretical limitations: ambiguities from degenerate eigenvalues and discontinuities

from axis flipping under perturbations. However, our empirical analysis shows that neither affects model performance in practice: principal-moment degeneracy affects only 0.007% of QM9 and is effectively zero on GEOM-Drugs; perturbation-induced instabilities are equally negligible (details in Appendix E.1). Moreover, we emphasize that our contribution is **not** to propose a theoretically perfect canonicalization method. Rather, we are the first to put inertial-frame canonicalization in service of autoregressive 3D molecule generation by turning it into a generation-oriented tokenization, showing that this simple strategy is sufficiently robust to achieve SOTA results (Appendix A).

**Step 2: Canonical Reordering of Atom Index**. To resolve the permutation ambiguity of atom indexing, we leverage the canonical ordering provided by RDKit (Landrum, 2016). Specifically, the reordering is determined by the serialization process of canonical SMILES generation, where atoms are ranked via an iterative invariant refinement procedure based on properties such as atomic number, connectivity, and ring membership. We then re-index atoms according to this order, reducing $n!$ possible permutations to one unique sequence for AR modeling.

### 3.2. GeoPE: Geometric Positional Encoding

After obtaining the canonical sequence of tokens, each atom-based token $a_i = (t_i, c_i)$ defined in Equation (3) must be effectively encoded into a latent representation suitable for Transformer modeling. This representation should capture both the discrete atom type $t_i$ and the continuous 3D coordinates $c_i = (x_i, y_i, z_i)$, ensuring that the self-attention mechanism can fully perceive and reason about the chemical identity and spatial arrangement of atoms.

**Atom Type Embedding**. For the discrete atom type $t_i$, we employ a learnable embedding layer to map this categorical feature into a continuous, high-dimensional vector:

$$z_i^{\text{type}} = \text{Embedding}(t_i). \tag{4}$$

**Geometric Positional Encoding (GeoPE).** To enable the self-attention mechanism to capture the relative spatial relationships between atoms, a geometry-aware encoding of the continuous 3D coordinates $c_i = (x_i, y_i, z_i)$ is essential. To this end, we introduce the geometric positional encoding tailored for 3D point-based tokens, as shown in Figure 1(b). GeoPE integrates (i) Manhattan Positional Encoding (ManhattanPE) for relative positional awareness along spatial axes, and (ii) Euclidean Positional Encoding (EuclideanPE) for efficient modeling of pairwise Euclidean distances through Nyström approximation.

**(i) ManhattanPE for Axis-wise Relative Geometry.** To make the self-attention mechanism geometry-aware, the positional encoding must ensure the inner product for absolute positions $c_i$ and $c_j$ depends solely on their relative positions,

$c_j - c_i$. This can be expressed as:

$$
\begin{aligned}
R_{c_i}^T R_{c_j} &= R_{x_i,y_i,z_i}^T R_{x_j,y_j,z_j} \\
&= R_{x_j-x_i,y_j-y_i,z_j-z_i} = R_{c_j-c_i}.
\end{aligned} \tag{5}
$$

Here, $R_{x,y,z}$ is the positional encoding function that maps 3D coordinates to their latent representation. This forces the attention scores to reflect the molecule's internal geometry, not its arbitrary global orientation. Then, inspired by Su (2021), we propose the Manhattan Positional Encoding (ManhattanPE) for atom-based tokens in Euclidean space. Here, $\theta_0 > 0$ is a frequency hyperparameter:

$$
R_{x,y,z}\boldsymbol{q} = \begin{bmatrix} q_0 \\ q_1 \\ q_2 \\ q_3 \\ q_4 \\ q_5 \end{bmatrix} \cdot \begin{bmatrix} \cos x\theta_0 \\ \cos x\theta_0 \\ \cos y\theta_0 \\ \cos y\theta_0 \\ \cos z\theta_0 \\ \cos z\theta_0 \end{bmatrix} + \begin{bmatrix} -q_1 \\ q_0 \\ -q_3 \\ q_2 \\ -q_5 \\ q_4 \end{bmatrix} \cdot \begin{bmatrix} \sin x\theta_0 \\ \sin x\theta_0 \\ \sin y\theta_0 \\ \sin y\theta_0 \\ \sin z\theta_0 \\ \sin z\theta_0 \end{bmatrix}. \tag{6}
$$

This ManhattanPE in Equation (6) is then applied to the query $\boldsymbol{q}$ and key $\boldsymbol{k}$ vectors of each atom within the self-attention mechanism. A crucial outcome of this formulation is that the inner product between a query vector transformed by position $c_i$ and a key vector transformed by position $c_j$ becomes a function of only their relative positions, $c_j - c_i$:

$$
\begin{aligned}
(R_{c_i}\boldsymbol{q})^T (R_{c_j}\boldsymbol{k}) &= (R_{x_i,y_i,z_i}\boldsymbol{q})^T (R_{x_j,y_j,z_j}\boldsymbol{k}) \\
&= \boldsymbol{q}^T R_{x_j-x_i,y_j-y_i,z_j-z_i}\boldsymbol{k} \\
&= \boldsymbol{q}^T R_{c_j-c_i}\boldsymbol{k}.
\end{aligned} \tag{7}
$$

Consequently, the attention score between any two atoms depends on their feature representations (via $\boldsymbol{q}$ and $\boldsymbol{k}$) and their relative spatial arrangement, fulfilling the initial requirement for a geometry-aware self-attention mechanism.

**(ii) EuclideanPE for Pairwise Distance Geometry.** One limitation of using ManhattanPE in Equation (6) is that it treats each axis separately; although it may implicitly capture pairwise distance information, we empirically observe that explicitly adding pairwise Euclidean distance information is more informative.

Then the question is how to explicitly incorporate the pairwise distance into the model. One straightforward way is to directly inject the distance information into the attention score, like (Shi et al., 2023). However, such an architecture is not compatible with the standard Transformer architecture used in large language models (Bai et al., 2023; Achiam et al., 2023; Touvron et al., 2023).

To alleviate this issue, we implement EuclideanPE with the Nyström method (Williams & Seeger, 2000), which provides a low-rank approximation of the RBF kernel defined over pairwise Euclidean distances. More concretely, suppose we have a Gram matrix over $n$ points,

*i.e.*, $K \in \mathbb{R}^{n \times n}$. Each element $K_{ij}$ is the radial basis function (RBF) over the distance between $i$-th and $j$-th points, $K_{ij} = RBF(c_i, c_j) = \exp(-\frac{\|c_i-c_j\|^2}{2\sigma^2})$, with $c_i$ denoting the 3D coordinates of the $i$-th point in Euclidean space. Then we sample $m$ anchor points, $(\tilde{c}_1, \tilde{c}_2, \ldots, \tilde{c}_m)$ with $m \ll n$. The RBF kernel over these $m$ anchor points forms an $m \times m$ matrix $A \in \mathbb{R}^{m \times m}$ with positive eigenvalues. By Cholesky decomposition, we have $A = LL^T$. Then, to approximate $K_{ij}$, we first construct the feature vector between point $i$ and the $m$ anchor points as $k_i = [RBF(c_i, \tilde{c}_1), RBF(c_i, \tilde{c}_2), \ldots, RBF(c_i, \tilde{c}_m)]^T \in \mathbb{R}^{m \times 1}$. For each atom $i$, we define its Euclidean positional encoding (abbreviated as $z_i^{\text{Euc}}$) as:

$$
z_i^{\text{Euc}} = L^{-1} k_i. \tag{8}
$$

This allows the approximated RBF, which encodes the pairwise distance information between atoms, to be recovered directly by the inner product in the attention mechanism (details are in Appendix C):

$$
\tilde{k}(i, j) = (z_i^{\text{Euc}})^T (z_j^{\text{Euc}}). \tag{9}
$$

**Latent Representation of Atom-based Token**. The final input representation for each atom $i$ is the concatenation of its type embedding and its Euclidean positional encoding:

$$
z_i = [z_i^{\text{type}}, z_i^{\text{Euc}}]. \tag{10}
$$

Within the attention layer, the input representation $z_i$ is projected into query $q_i$, key $k_i$, and value $v_i$. Here, we take the query projection for illustration:

$$
q_i = W_q z_i. \tag{11}
$$

Crucially, to maintain the distinct roles of the atom type embedding and Euclidean positional encoding, the weight matrix $W_q$ is structured as a block-diagonal matrix. This structure ensures that the two components of the input representation are projected independently. Recall that $z_i = [z_i^{\text{type}}, z_i^{\text{Euc}}]$, the projection is implemented as:

$$
\begin{bmatrix} q_i^{\text{type}} \\ q_i^{\text{Euc}} \end{bmatrix} = \begin{bmatrix} W_q^{\text{type}} & 0 \\ 0 & W_q^{\text{Euc}} \end{bmatrix} \begin{bmatrix} z_i^{\text{type}} \\ z_i^{\text{Euc}} \end{bmatrix}, \tag{12}
$$

where $W_q^{\text{type}}$ is the learnable weight matrix for the type component, and $W_q^{\text{Euc}}$ is the identity matrix. The key $k_i$ and value $v_i$ are computed in an analogous manner using their own block-diagonal weight matrices, $W_k$ and $W_v$. ManhattanPE is applied only to the type-derived query and key components, while EuclideanPE remains as a separate feature branch. The final query and key vectors are formed by concatenating these two parts, where we use the superscript Man (short for Manhattan) to denote the

ManhattanPE-transformed components: $q_i^{\text{Man}} = R_{c_i} q_i^{\text{type}}$ and $k_j^{\text{Man}} = R_{c_j} k_j^{\text{type}}$.

$$\tilde{q}_i = \begin{bmatrix} R_{c_i} q_i^{\text{type}} \\ q_i^{\text{Euc}} \end{bmatrix} = \begin{bmatrix} q_i^{\text{Man}} \\ q_i^{\text{Euc}} \end{bmatrix}. \tag{13}$$

$$\tilde{k}_j = \begin{bmatrix} R_{c_j} k_j^{\text{type}} \\ k_j^{\text{Euc}} \end{bmatrix} = \begin{bmatrix} k_j^{\text{Man}} \\ k_j^{\text{Euc}} \end{bmatrix}. \tag{14}$$

The key advantage of this construction is revealed in the inner product, which combines the two sources of geometric information. The attention score between atoms $i$ and $j$ is computed as:

$$\begin{aligned}
\text{Attn}(i,j) &= \tilde{q}_i^T \tilde{k}_j \\
&= (q_i^{\text{Man}})^T k_j^{\text{Man}} + (q_i^{\text{Euc}})^T (k_j^{\text{Euc}}) \\
&= (q_i^{\text{type}})^T \underbrace{R_{c_j - c_i} k_j^{\text{type}}}_{\substack{\text{Manhattan} \\ \text{Distance}}} + \underbrace{\text{RBF}(\|c_i - c_j\|)}_{\substack{\text{Euclidean Distance} \\ \text{(Nyström)}}}.
\end{aligned} \tag{15}$$

This formulation ensures that the self-attention score explicitly and simultaneously models both Manhattan-style relative geometry via ManhattanPE and Euclidean pairwise geometry via EuclideanPE, providing a rich and robust inductive bias.

### 3.3. Hierarchical Autoregressive Architecture

The sequence of latent representations derived from Section 3.2, $(z_1, \ldots, z_n)$, is then processed by the autoregressive Transformer backbone to produce a sequence of context-aware hidden embeddings, $(h_1, \ldots, h_n)$. At each step $i$, the hidden embedding $h_i$, which encapsulates the full context of the previous atoms $a_{<i+1}$, is used to predict the next token, $a_{i+1} = (t_{i+1}, c_{i+1})$. This presents a unique challenge, as the prediction target is a hybrid of a discrete type and a continuous coordinate vector. To address this, we factorize the conditional probability into two components:

$$\begin{aligned}
p(a_{i+1} \mid h_i) &= p(t_{i+1}, c_{i+1} \mid h_i) \\
&= \underbrace{p(t_{i+1} \mid h_i)}_{\text{Type}} \cdot \underbrace{p(c_{i+1} \mid t_{i+1}, h_i)}_{\text{3D Coordinates}}.
\end{aligned} \tag{16}$$

In Equation (16), the model first predicts the atom type $t_{i+1}$ conditioned on the hidden embedding $h_i$. Subsequently, the continuous 3D coordinates $c_{i+1}$ are predicted given both $t_{i+1}$ and $h_i$. Concretely, we implement this using a hierarchical AR architecture (as illustrated in Figure 1(c)): (i) a type-prediction block dedicated to modeling the discrete, categorical distribution over atom types, and (ii) a coordinates-prediction block to predict continuous 3D coordinates. This hierarchical architecture not only aligns with the intrinsic nature of molecular generation but also enhances learning efficiency by decoupling the tasks of categorical classification and continuous density estimation (Cheng et al., 2025b).

**Cross-Entropy Loss for Type Prediction Block.** For the discrete atom type $t_{i+1}$, we employ the standard cross-entropy, which directly maximizes the likelihood of the ground-truth atom type given the hidden embedding $h_i$:

$$\mathcal{L}_{\text{type}} = -\mathbb{E}_{(h_i, t_{i+1}) \sim \mathcal{D}} \left[ \log p_\theta(t_{i+1} \mid h_i) \right]. \tag{17}$$

**Diffusion Loss for 3D Coordinates Prediction Block.** Autoregressive models are naturally well-suited for generating discrete tokens using cross-entropy. However, for continuous 3D coordinates $c_{i+1}$, we empirically find that direct regression yields poor performance. To overcome this limitation, we adopt Diffusion Loss from Li et al. (2024a), which provides an effective framework for extending autoregressive models to continuous-valued token generation. The high-level idea is that we perturb the ground-truth position $c_{i+1}$ by adding Gaussian noise with a sampled noise level $\sigma$, and train a denoising network $\epsilon_\theta$ to recover the injected noise (Karras et al., 2022). Concretely, the perturbed coordinate is given by

$$c_{i+1}^{(\sigma)} = c_{i+1} + \sigma \epsilon, \quad \epsilon \sim \mathcal{N}(0, I). \tag{18}$$

Conditioned on the hidden embedding $h_i$ and the predicted atom type $t_{i+1}$, the denoising network is optimized with the following loss function:

$$\mathcal{L}_{\text{diff}} = \mathbb{E}_{\sigma, c_{i+1}, \epsilon} \left[ \left\| \epsilon - \epsilon_\theta(c_{i+1}^{(\sigma)}, \sigma, t_{i+1}, h_i) \right\|_2^2 \right]. \tag{19}$$

This objective enables the coordinates prediction block to model the continuous distribution of atom positions. At inference time, atom coordinates are generated by iterative denoising from Gaussian noise, conditioned on both the autoregressive context $h_i$ and the sampled atom type $t_{i+1}$.

**Controllable Generation with Classifier-free Guidance.** We incorporate classifier-free guidance (CFG) into InertialAR to enable controllable generation. During inference, CFG combines the conditional and unconditional predictions as:

$$p_g = p_u + s(p_c - p_u),$$

where $p_c$ and $p_u$ denote the conditional and unconditional predictions, respectively, and $s$ is the guidance scale. In InertialAR, CFG is applied to both the diffusion noise prediction for coordinate generation and the pre-softmax logits for atom type prediction by combining conditional and unconditional outputs with a guidance scale $s$. By tuning $s$, we can achieve both stronger adherence to target molecular classes and better structural validity.

## 4. Experiments

### 4.1. Unconditional 3D Molecule Generation

**QM9 and GEOM-Drugs Dataset.** We use QM9 (Ramakrishnan et al., 2014) and GEOM-Drugs (Axelrod &

*Table 1.* Unconditional molecule generation results on QM9 and GEOM-Drugs. Best results are **bolded**. Results for MiDi are marked with [†] to indicate re-evaluation using the standardized protocol from EDM for fair comparison.

| Method | Backbone | QM9 | | | | GEOM-Drugs | |
|---|---|---|---|---|---|---|---|
| | | Valid (%) | Valid&Uni (%) | AtomSta (%) | MolSta (%) | Valid (%) | AtomSta (%) |
| E-NFs | GNN | 40.2 | 39.4 | 85.0 | 4.9 | – | – |
| G-SchNet | GNN | 85.5 | 80.3 | 95.7 | 68.1 | – | – |
| GDM | GNN | – | – | 97.0 | 63.2 | 90.8 | 75.0 |
| GDM-AUG | GNN | 90.4 | 89.5 | 97.6 | 71.6 | 91.8 | 77.7 |
| EDM | GNN | 91.9 | 90.7 | 98.7 | 82.0 | 92.6 | 81.3 |
| EDM-Bridge | GNN | 92.0 | 90.7 | 98.8 | 84.6 | 92.8 | 82.4 |
| MiDi[†] | GNN | 94.2 | 92.9 | 98.2 | 83.8 | 92.1 | 75.6 |
| GeoLDM | GNN | 93.8 | 92.7 | 98.9 | 89.4 | **99.3** | 84.4 |
| UniGEM | GNN | 95.0 | **93.2** | 99.0 | 89.8 | 98.4 | 85.1 |
| Geo2Seq | Transformer | 97.1 | 81.7 | 98.9 | 93.2 | 96.1 | 82.5 |
| InertialAR (Ours) | Transformer | **97.3** | 92.5 | **99.2** | **94.5** | 98.0 | **87.0** |

*Table 2.* Unconditional generation results on B3LYP-1M.

| Model | Valid (%) | Valid&Uni (%) | AtomSta (%) | MolSta (%) |
|---|---|---|---|---|
| EDM | 92.9 | 92.8 | 80.6 | 0.8 |
| Geo2Seq | 73.3 | 2.7 | 10.0 | 0.0 |
| InertialAR | **99.3** | **98.8** | **84.8** | **24.7** |

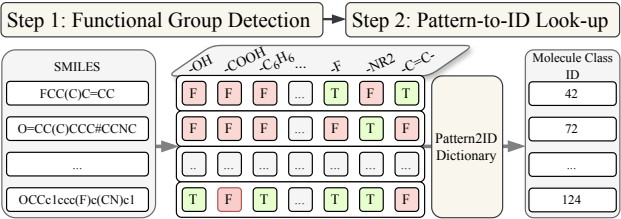

*Figure 3.* Mapping 3D molecules to Molecule Class IDs.

Gómez-Bombarelli, 2022) for unconditional 3D molecular generation. QM9 contains 130K small molecules with high-quality 3D conformations (up to 9 heavy atoms). We split the dataset into train, validation and test sets with 100K, 17K and 13K samples, respectively. GEOM-Drugs consists of 37M conformations for around 450K unique molecules (up to 181 atoms and 44.2 atoms on average). Following Hoogeboom et al. (2022), we select the 30 lowest-energy conformations per molecule for training.

**B3LYP Dataset.** Moreover, we evaluate on a brand new, larger, and more comprehensive 3D molecular dataset, the PubChemQC B3LYP/6-31G//PM6 dataset (abbreviated as B3LYP) (Nakata & Maeda, 2023). This dataset contains a total of 85,938,443 molecules, covering a wide range of chemical diversity with molecular weights up to 1000 and more than 50 different atom types. We use a subset of 1M molecules for training. The evaluation metrics remain consistent with those used for QM9 and GEOM-Drugs.

**Evaluation.** Model performance is assessed through a set of chemical feasibility metrics. Bond types (single, double, triple, or none) are determined from molecular geometries based on pairwise atomic distances and atom identities. The evaluation includes Atom Stability (proportion of atoms satisfying correct valency), Molecule Stability (proportion of molecules in which all atoms are stable), Validity (fraction of chemically valid molecules as verified by RDKit), and Uniqueness (fraction of non-duplicate molecules among generated samples). All metrics are computed following evaluation protocols in Hoogeboom et al. (2022).

**Baselines.** We benchmark InertialAR against G-SchNet (Gebauer et al., 2019), E-NFs (Satorras et al., 2022a), EDM (Hoogeboom et al., 2022), GDM (Hoogeboom et al., 2022), EDM-Bridge (Wu et al., 2022), MiDi (Vignac et al., 2023), GeoLDM (Xu et al., 2023), UniGEM (Feng et al., 2025) and Geo2Seq (Li et al., 2024b).

**Results on QM9 and GEOM-Drugs.** Table 1 highlights the strong performance of InertialAR across both QM9 and GEOM-Drugs benchmarks. On QM9, InertialAR achieves the highest scores on Valid, Atom Stability and Molecule Stability, surpassing all competing methods and indicating its ability to generate chemically consistent and structurally reliable molecules. On the larger and more complex GEOM-Drugs dataset, InertialAR continues to demonstrate superiority, attaining the best Atom Stability among all baselines. These results underscore the robustness of InertialAR in ensuring both chemical validity and structural stability, validating its effectiveness as a powerful autoregressive framework for 3D molecule generation.

**Results on B3LYP.** Due to the prohibitive computational cost of training all existing models on the large-scale B3LYP benchmark, we focus our comparison on two representative strong baselines: the diffusion-based EDM and the autoregressive Geo2Seq. The main results are shown in Table 2. InertialAR achieves substantial improvements over base-

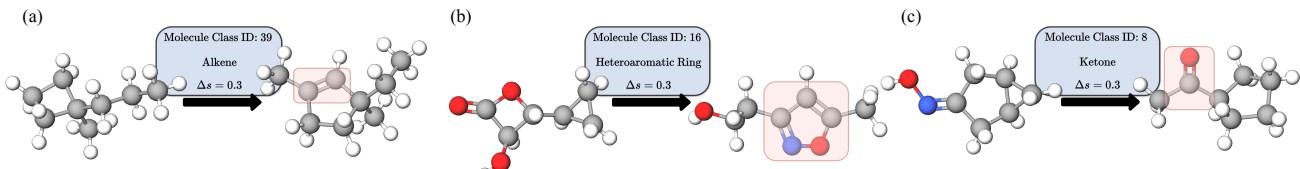

*Figure 4.* Visualization of molecule editing by tuning the CFG guidance scale $s$.

*Table 3.* Class-conditional generation results on QM9. We report hit rate, validity, uniqueness, and stability metrics across different functional group targets (Molecule Class IDs).

| Model | Hit Rate (%) | Valid (%) | Valid&Uni (%) | AtomSta (%) | MolSta (%) |
|---|---|---|---|---|---|
| *Class 7: w/ Ether* | | | | | |
| EDM | 37.5 | 84.8 | 84.2 | 96.3 | 52.9 |
| Geo2Seq | 40.1 | 65.0 | 52.1 | 87.6 | 33.8 |
| InertialAR | **90.9** | **99.0** | **92.8** | **99.7** | **97.5** |
| *Class 28: w/ Hydroxyl & Ether* | | | | | |
| EDM | 29.0 | 86.8 | 85.9 | 96.4 | 54.1 |
| Geo2Seq | 44.2 | 64.7 | 55.6 | 86.5 | 33.4 |
| InertialAR | **89.8** | **99.9** | **90.8** | **99.9** | **99.2** |
| *Class 3: w/ Hydroxyl* | | | | | |
| EDM | 27.6 | 85.3 | 84.0 | 96.7 | 56.5 |
| Geo2Seq | 49.4 | 70.3 | 53.9 | 89.7 | 42.2 |
| InertialAR | **85.7** | **99.9** | **86.9** | **99.9** | **99.4** |
| *Class 16: w/ Heteroaromatic Ring* | | | | | |
| EDM | 8.9 | 63.5 | 63.4 | 82.9 | 35.3 |
| Geo2Seq | 33.8 | 65.6 | 57.8 | 86.4 | 34.8 |
| InertialAR | **68.5** | **92.2** | **79.3** | **97.1** | **81.0** |
| *Class 23: w/ Secondary Amine & Ether* | | | | | |
| EDM | 25.3 | 76.8 | 76.7 | 96.1 | 53.3 |
| Geo2Seq | 43.5 | 80.5 | 51.7 | 91.8 | 52.4 |
| InertialAR | **81.8** | **99.7** | **82.7** | **99.9** | **99.2** |

lines on the large-scale B3LYP benchmark and attains the best results across all four metrics. Compared to the strong diffusion model EDM, it achieves significantly higher validity and atom stability. Most notably, InertialAR shows a dramatic gain in Molecule Stability, demonstrating its ability to produce chemically consistent molecules at scale. In contrast, the autoregressive baseline Geo2Seq performs poorly, highlighting the robustness and scalability of our approach on this chemically diverse dataset.

**Ablations.** We further conduct component-wise ablations to validate the effectiveness of our architecture design choices. In particular, GeoPE is crucial for autoregressive 3D molecule generation, and both ManhattanPE and EuclideanPE contribute meaningfully (Appendix E.2). Moreover, replacing Diffusion Loss with direct L2 regression causes a sharp degradation in generation quality (Appendix E.3), and removing deterministic atom reordering also consistently harms validity and uniqueness (Appendix E.4).

### 4.2. Class-conditional 3D Molecule Generation and Molecule Editing

In chemistry and biology, class-conditional generation is particularly valuable, as "molecule classes" can correspond to key attributes such as chemical functionality, thereby enabling the targeted design or editing of molecules for drug discovery and materials science.

To enable class-conditional generation on QM9, we reconstruct the dataset by assigning each molecule a **Molecule Class ID** that encodes its functional group configuration (as shown in Figure 3). Specifically, we first convert each 3D structure to its SMILES string and then apply a rule-based SMARTS-matching system to detect predefined functional groups. The resulting presence/absence pattern is encoded as a binary string (*e.g.* "TFTT..."). Finally, through a predefined Functional Group Pattern-to-Class ID look-up, each molecule is assigned a corresponding Molecule Class ID.

The task is then to generate molecules conditioned on a specified functional group configuration. Concretely, we select the 5 most frequent Molecule Class IDs as conditioning targets. In addition to the metrics used for unconditional generation, we introduce a critical new metric for class-conditional generation, **Hit Rate**, which measures the proportion of generated molecules satisfying the target functional group requirements. A higher hit rate indicates stronger controllability of the generation process.

**Baselines.** We compare the conditional generation performance of InertialAR against the same representative autoregressive and diffusion-based baselines as in the unconditional setting, namely Geo2Seq and EDM, to ensure a consistent and fair comparison. **Results.** Table 3 shows that InertialAR achieves a remarkable average hit rate of 83.3%, significantly surpassing EDM and Geo2Seq, demonstrating its strong controllability in generating molecules that match the target functional group configurations. Beyond controllability, InertialAR also achieves excellent performance on chemical feasibility metrics, consistently outperforming both baselines across all evaluated molecule classes. These results highlight the effectiveness of InertialAR in producing both chemically valid and functionally precise molecules.

**Molecule Editing via CFG.** To further assess controllability, we examine the effect of varying the CFG guidance scale. Increasing the scale not only improves validity-related metrics but also enables molecule editing: molecules that

originally lacked the required functional groups and exhibited unreasonable structures can be transformed to satisfy the target Molecule Class ID. As illustrated in Figure 4, by raising the guidance scale by 0.3 ($\Delta s = 0.3$), the generated molecules incorporate the desired functional groups while yielding more plausible 3D geometries, demonstrating that CFG enhances both structural validity and compliance with functional group constraints.

**Discussion and Limitations.** Inertial-frame canonicalization is not theoretically unique in rare edge cases, such as exact principal-moment degeneracy or near-symmetric geometries. Nevertheless, Appendix E.1 shows that such cases are statistically negligible on QM9 and GEOM-Drugs, supporting the practical robustness of the proposed tokenization pipeline on real molecular data. Appendix E.5 further reports additional multi-seed stability results, which show that the overall performance trends remain stable across runs. To complement validity and stability metrics, Appendix E.6 reports RDKit synthetic accessibility (SA) scores for unconditional and class-conditional generation on QM9. The conditional scores remain broadly comparable to the unconditional baseline overall, suggesting no severe degradation in this proxy under CFG-driven control. However, SA is only a graph-level proxy and does not constitute a complete measure of synthesizability.

## 5. Conclusion

We propose InertialAR, a hierarchical autoregressive model for 3D molecule generation. First, InertialAR performs generation-oriented canonical tokenization by aligning each molecule to a canonical inertial frame and deterministically reordering atoms, thereby converting arbitrary 3D structures into a unique, SE(3)- and permutation-invariant sequence of tokens for autoregressive generation. Built upon this canonical tokenization, GeoPE equips Transformer attention with geometric awareness through ManhattanPE and EuclideanPE. Finally, InertialAR adopts a hierarchical autoregressive paradigm that predicts atom types via cross-entropy and 3D coordinates via Diffusion Loss.

**Future Directions.** Our results suggest that InertialAR can advance 3D molecule generation beyond restrictive physical priors and may serve as a foundation model component for scientific discovery. Looking ahead, InertialAR can be extended to more complex domains such as protein structure modeling and periodic material discovery, and can be integrated into broader multimodal frameworks.

## Impact Statement

This paper presents a novel framework for autoregressive 3D molecule generation, aiming to advance the technical capabilities of generative modeling in scientific domains. By enabling efficient exploration of chemical space, this work has the potential to accelerate research in drug discovery and materials science. However, as with any generative technology, there is a risk of misuse, such as the design of illicit or hazardous compounds. We emphasize that our method generates theoretical structures without synthesis protocols; practical deployment must adhere to strict safety screenings, ethical guidelines, and regulatory standards to mitigate potential harms.

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

# A. Extended Related Work

## A.1. 3D Molecule Generation

In the domain of AI-driven molecule discovery, 3D molecule generation has become a central problem. Its goal is to directly construct physically plausible 3D molecular conformations. Formally, a 3D molecule with $n$ atoms can be represented as a point cloud $\mathcal{M} = (t, C)$. The vector $t = [t_1, \cdots, t_n] \in \mathbb{Z}^n$ encodes the atom types (atomic numbers), and the coordinate matrix $C = [c_1, \cdots, c_n] \in \mathbb{R}^{3 \times n}$ specifies the 3D position of each atom, with $c_i \in \mathbb{R}^3$. A fundamental challenge lies in ensuring that molecular geometries respect the inherent SE(3) symmetry, i.e., molecular representations must remain invariant or equivariant under SE(3) transformations such as rotations and translations.

Current approaches can be categorized into four main paradigms. SE(3)-equivariant architectures explicitly enforce symmetry through specialized network designs: spherical frame basis models (Thomas et al., 2018; Liao & Smidt, 2023) project features into irreducible representations of SO(3), while vector frame basis models (Satorras et al., 2022b; Schütt et al., 2021) construct local coordinate frames for equivariant operations. Invariant feature approaches circumvent architectural constraints by utilizing geometrically invariant inputs such as pairwise distances, bond angles, and dihedral angles (Schütt et al., 2017). Data augmentation strategies encourage models to implicitly learn symmetric representations by training on randomly rotated and translated molecular conformations, particularly valuable for large-scale models where explicit equivariance is complex to scale (Abramson et al., 2024). Input canonicalization methods (Li et al., 2024b; Fu et al., 2024) establish a canonical orientation or reference frame for input molecules through preprocessing, transforming each molecule into a standardized pose so that subsequent neural networks can operate on SE(3)-invariant inputs without intrinsic SE(3)-equivariant constraints.

A representative canonicalization strategy defines an inertial reference frame for each molecule using principal component analysis (PCA) (Guo et al., 2025; Lu et al., 2025a; Cheng et al., 2025a). Similar ideas have been widely explored in biology (Zhou et al., 2026; Ni et al., 2026). After shifting the molecular coordinates so that the center of mass lies at the origin, the moment of inertia matrix is diagonalized to obtain the principal axes of rotation. Aligning the coordinates with these axes yields a canonical pose, unique up to axis reflections, effectively removing translational and rotational ambiguities. This inertial frame ensures SE(3)-symmetry molecular representations, enabling neural networks to process standardized and physically consistent 3D geometries without explicit equivariant design.

## A.2. Inertial Frame and PCA

PCA-based inertial frames provide a simple and effective practical canonicalization strategy. Empirically, we find that PCA canonical poses are highly stable on real molecular datasets, making them an efficient SE(3) canonicalization choice for unconstrained architectures (details in Appendix E). Theoretically, however, PCA-based canonicalization is not strictly unique. Its limitations include potential axis flips from small geometric perturbations and ambiguity in axis orientation when principal moments are tied. These theoretical non-uniqueness issues have motivated a line of canonicalization-based symmetry handling methods that study how to systematically manage symmetry-equivalent frames. Frame Averaging (Puny et al., 2022) treats canonicalization as an equivariant projection by averaging outputs across all symmetry-equivalent PCA frames, while subsequent work shows that any finite, unweighted canonicalization procedure necessarily introduces discontinuities under symmetric configurations (Dym et al., 2024). Our approach is complementary to this line: we adopt our canonical inertial frame as a simple and empirically robust canonicalization strategy, while these canonicalization-based methods provide principled tools that could further enhance robustness in future extensions.

## A.3. Autoregressive Models and Tokenization of 3D Molecules

Autoregressive models address sequence modeling by framing it as a "next-token prediction" problem. This approach, a direct application of the chain rule of probability, factorizes the joint distribution of a sequence $x = (x_1, \ldots, x_n)$ into a product of conditional probabilities:

$$p(x) = p(x_1, \ldots, x_n) = \prod_{i=1}^{n} p(x_i | x_1, \ldots, x_{i-1}). \tag{20}$$

The model's core task is thus to learn the conditional distribution $p(x_i | x_{<i})$ for each step, which is typically parameterized by a powerful neural network such as Transformer. The primary challenge in applying autoregressive models to 3D molecular generation lies in the effective **structure tokenization** of a 3D molecular structure into a 1D sequence of tokens suitable for

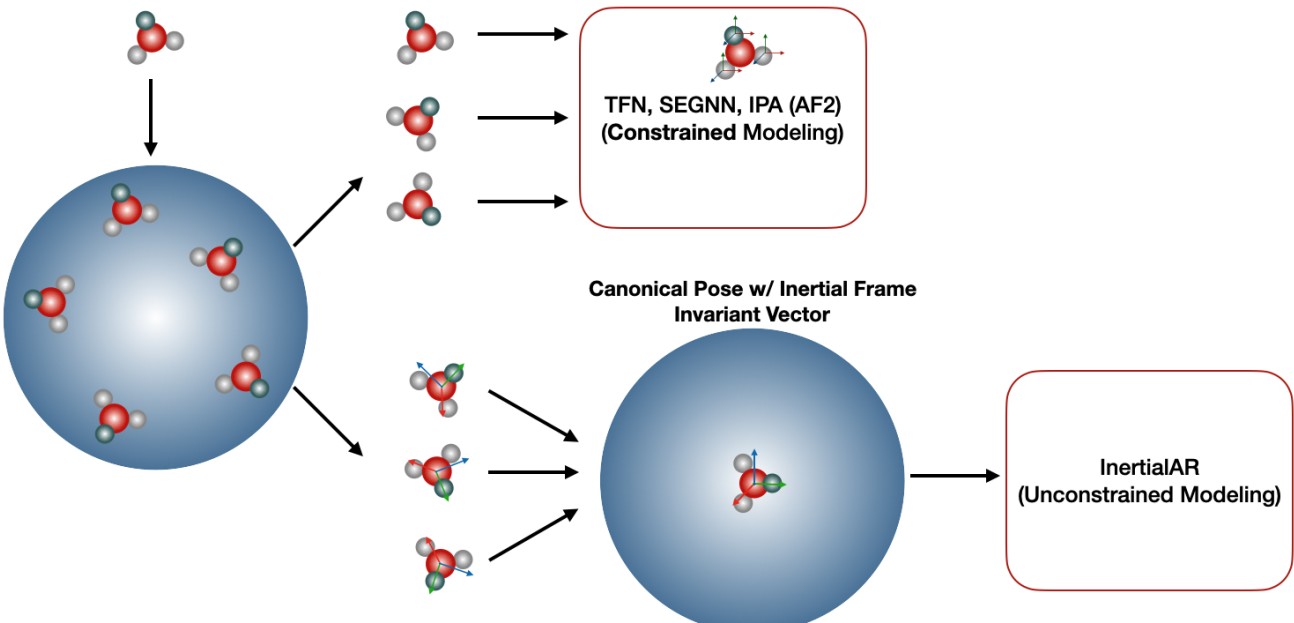

*Figure 5.* Comparison of existing SE(3)-equivariant graph neural networks and InertialAR.

Transformer architectures. The choice of tokenization strategy is crucial, as it defines not only the sequence representation but also the very nature of the conditional modeling itself. Existing approaches can be broadly classified into three main categories:

**Voxel-based tokenization**, which discretizes the 3D space occupied by a molecule into a 3D grid, draws a direct parallel to image generation (Faltings et al., 2025; Lu et al., 2025b). Each voxel in the grid serves as a token that encodes local atomic information, much like a pixel in an image. **Text sequence-based tokenization**, which is similar to language modeling, serializes 3D molecules into a 1D, text-like sequence (Li et al., 2024b; Yan et al., 2024; Flam-Shepherd & Aspuru-Guzik, 2023). The process involves discretizing continuous 3D coordinates and concatenating them with discrete atom types. This treats a molecule like a sentence, where every atom type and 3D coordinates are encoded as words. **Atom-based tokenization** directly treats an atom as one single token that encapsulates both its discrete atom type and continuous 3D coordinates. This establishes an intuitive correspondence between the physical atoms and their tokenized representation, thereby preserving atom-level granularity.

### A.4. Class-conditional Generation and Classifier-free Guidance

Class-conditional generation is a paradigm that generates samples conditioned on a specific class label $c$. In image generation, this involves generating an image guided by a prefix class embedding (Esser et al., 2021; Peebles & Xie, 2023). In chemistry and biology, class-conditional generation is highly useful, as molecular "classes" can correspond to key attributes such as chemical functionality or physicochemical characteristics, enabling the targeted design or editing of molecules for drug discovery and materials science.

Classifier-free guidance (CFG) improves both sample quality and fidelity to conditions by randomly dropping conditioning signals during training (Ho & Salimans, 2022). This simple yet effective strategy enables a single model to jointly learn both the conditional distribution $p(x|c)$ and the unconditional distribution $p(x)$. At inference, the difference between these two learned distributions is then leveraged to amplify the conditional signal without relying on an auxiliary classifier. Although originally proposed for diffusion, CFG has also proven effective in autoregressive image generation, showing great potential for molecule generation.

### A.5. Diffusion Loss for Autoregressive Models

While autoregressive models are naturally suited for generating discrete tokens via cross-entropy loss, 3D molecule generation introduces an additional challenge: predicting continuous 3D coordinates. Diffusion Loss (Li et al., 2024a)

provides an effective framework to extend autoregressive models to continuous-valued token generation. Formally, to predict the continuous-valued token $x_i$, the autoregressive model first outputs a vector $h_{i-1}$ conditioned on previous tokens $x_{<i}$. The objective is to model the conditional probability distribution $p(x_i|h_{i-1})$. Diffusion loss achieves this through a denoising score-matching objective:

$$L(x_i, h_{i-1}) = \mathbb{E}_{\epsilon,t} \left[ |\epsilon - \epsilon_\theta(x_i^t|t, h_{i-1})|^2 \right], \tag{21}$$

where $x_i^t = \sqrt{\bar{\alpha}_t} x_i + \sqrt{1 - \bar{\alpha}_t} \epsilon$ is a noised version of $x_i$, and $\epsilon_\theta$ is a denoising network that predicts the noise $\epsilon$ conditioned on $h_{i-1}$ and timestep $t$. Gradients from this loss propagate through $h_{i-1}$, enabling end-to-end training of the autoregressive backbone.

This approach preserves the strong sequence modeling capacity of autoregressive models while extending them to predict continuous distributions. By directly modeling 3D coordinates, it removes the need for discretization or coarse tokenization of molecular geometries and provides a principled mechanism for generating chemically precise molecular structures.

# B. Class-conditional Generation

## B.1. Dataset Reconstruction

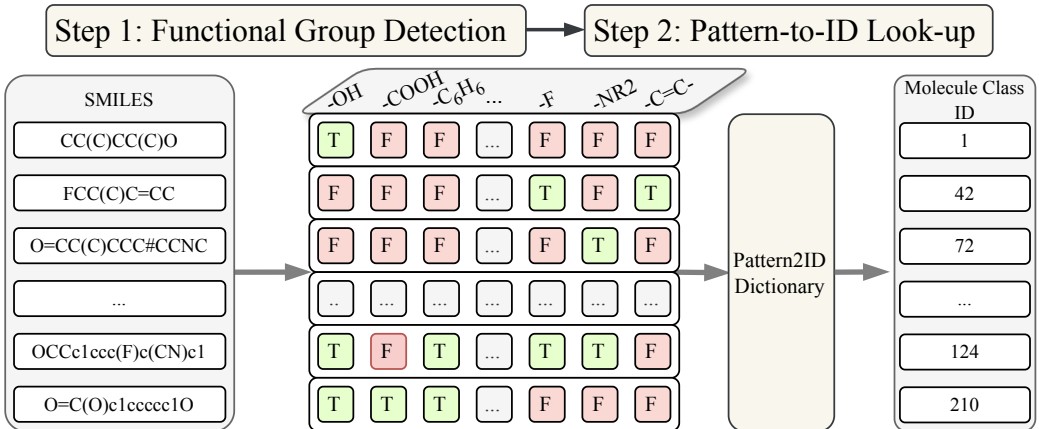

*Figure 6.* Overview of how 3D molecules are mapped to their Molecule Class IDs.

In chemistry and biology, class-conditional generation is highly useful, as "molecule classes" can correspond to key attributes such as chemical functionality or physicochemical characteristics, enabling the targeted design or editing of molecules for drug discovery and materials science. However, commonly used datasets, such as QM9 and GEOM-Drugs, do not provide explicit functional group annotations. To enable controllable molecule generation with specified functional group configurations, we reconstruct the datasets by assigning each molecule a unique class label (Molecule Class ID) that encodes its functional group composition. Concretely, we design a comprehensive labeling pipeline based on functional groups (shown in Figure 6): for each molecule, we first convert its 3D structure to a SMILES representation. We then employ a rule-based system with a library of SMARTS queries to identify the presence or absence of a predefined set of functional groups. The resulting pattern is encoded as a binary string (e.g., "TTFFTFTT..."), where each position indicates the presence (T) or absence (F) of a functional group. Finally, through a predefined Functional Group Pattern-to-Class ID mapping, each molecule is assigned a corresponding Molecule Class ID.

## B.2. Controllable Generation with Classifier-free Guidance

Originally developed in the diffusion model community, classifier-free guidance (CFG) is widely recognized for improving both sample quality and conditional alignment. The key idea is to train a single model that jointly learns the conditional distribution $p(x|c)$ and the unconditional distribution $p(x)$ by randomly dropping conditioning labels during training.

We adopt CFG in InertialAR following Section 2. In InertialAR, CFG is applied to the estimated noise $\epsilon_\theta$ in diffusion for coordinates generation, as well as to the logits over a discrete vocabulary for atom type prediction. By tuning the guidance scale, we can achieve both stronger adherence to target molecular classes and better structural validity.

## C. Nyström Estimation for EuclideanPE

The Nyström method (Williams & Seeger, 2000) provides a low-rank approximation of the RBF kernel defined over pairwise Euclidean distances. More concretely, suppose we have a Gram matrix over $n$ points, *i.e.*, $K \in \mathbb{R}^{n \times n}$. Each element $K_{ij}$ is the radial basis function (RBF) over the distance between the $i$-th and $j$-th points, $K_{ij} = \text{RBF}(c_i, c_j) = \exp\left(-\frac{\|c_i - c_j\|^2}{2\sigma^2}\right)$, with $c_i$ denoting the 3D coordinates of the $i$-th point. We sample $m$ anchor points, denoted by $(\tilde{c}_1, \tilde{c}_2, \ldots, \tilde{c}_m)$ with $m \ll n$, and permute $K$ accordingly so that the anchors correspond to the first $m$ indices.

First, we can decompose the matrix $K$ with eigendecomposition,

$$K = U\Lambda U^T, \tag{22}$$

where $U \in \mathbb{R}^{n \times n}$ is an orthogonal matrix whose columns are the orthonormal eigenvectors of $K$, and $\Lambda \in \mathbb{R}^{n \times n}$ is a diagonal matrix whose entries are the corresponding eigenvalues of $K$.

Then, Nyström approximation is a low rank approximation, assuming that matrix $K$ can be approximated using $\tilde{K}$:

$$\begin{aligned}
K &\approx \tilde{K} \\
&= \tilde{U}\tilde{\Lambda}\tilde{U}^T \\
&= \begin{bmatrix} A & B \\ B^T & C \end{bmatrix},
\end{aligned} \tag{23}$$

where $\tilde{U}$ is the first $m$ columns of $U$ and $\tilde{\Lambda}$ is the diagonal matrix of the first $m$ eigenvalues of $\Lambda$. At this point, we assume that the $m$ points picked can estimate the $m \times m$ matrix $A$ with positive eigenvalues. Then let us have $\tilde{U} = \begin{bmatrix} \tilde{U}_1 \\ \tilde{U}_2 \end{bmatrix}$, where $\tilde{U}_1 \in \mathbb{R}^{m \times m}$ and $\tilde{U}_2 \in \mathbb{R}^{(n-m) \times m}$. This means $A = \tilde{U}_1 \tilde{\Lambda} \tilde{U}_1^T$ and $B = \tilde{U}_1 \tilde{\Lambda} \tilde{U}_2^T$. Thus, we can rewrite Equation (23) as:

$$\begin{aligned}
\tilde{K} &= \begin{bmatrix} \tilde{U}_1 \\ \tilde{U}_2 \end{bmatrix} \tilde{\Lambda} \begin{bmatrix} \tilde{U}_1 \\ \tilde{U}_2 \end{bmatrix}^T \\
&= \begin{bmatrix} \tilde{U}_1 \tilde{\Lambda} \tilde{U}_1^T & \tilde{U}_1 \tilde{\Lambda} \tilde{U}_2^T \\ \tilde{U}_2 \tilde{\Lambda} \tilde{U}_1^T & \tilde{U}_2 \tilde{\Lambda} \tilde{U}_2^T \end{bmatrix}.
\end{aligned} \tag{24}$$

Combining this with Equation (23), we have $\tilde{U}_2 = B^T \tilde{U}_1 \tilde{\Lambda}^{-1}$ and $\tilde{U}_2^T = \tilde{\Lambda}^{-1} \tilde{U}_1^T B$. Thus, we can have

$$C = \tilde{U}_2 \tilde{\Lambda} \tilde{U}_2^T = B^T \tilde{U}_1 \tilde{\Lambda}^{-1} \tilde{U}_1^T B = B^T A^{-1} B. \tag{25}$$

To inject this back to Equation (23), we have

$$\begin{aligned}
\tilde{K} &= \begin{bmatrix} A & B \\ B^T & B^T A^{-1} B \end{bmatrix} \\
&= \begin{bmatrix} A \\ B^T \end{bmatrix} A^{-1} \begin{bmatrix} A & B \end{bmatrix}.
\end{aligned} \tag{26}$$

This wraps up the key idea of Nyström method. Then, to approximate $K_{ij}$, we first construct the feature between point $i$ and the $m$ anchor points as $k_i = [\text{RBF}(c_i, \tilde{c}_1), \text{RBF}(c_i, \tilde{c}_2), \ldots, \text{RBF}(c_i, \tilde{c}_m)]^T \in \mathbb{R}^{m \times 1}$. The approximated $\text{RBF}(i, j)$ can be obtained as:

$$\begin{aligned}
\tilde{k}(i, j) &= k_i^T A^{-1} k_j \\
&= \left(A^{-1/2} k_i\right)^T \left(A^{-1/2} k_j\right) \\
&= \left(L^{-1} k_i\right)^T \left(L^{-1} k_j\right),
\end{aligned} \tag{27}$$

where $A = LL^T$ is the Cholesky decomposition.

For each atom $i$, we define its Euclidean positional encoding as

$$z_i^{\text{Euc}} = L^{-1} k_i. \tag{28}$$

This design allows the approximated RBF, which encodes the pairwise distance information between atoms, to be recovered directly by the inner product within the attention mechanism:

$$\tilde{k}(i,j) = (z_i^{\text{Euc}})^T (z_j^{\text{Euc}}).$$
(29)

**Discussion.** There is another research line using random features (*e.g.*, random Fourier features) for the pairwise distance approximation (Rahimi & Recht, 2007). Certain works have shown that Nyström method can be more accurate (Yang et al., 2012). One intuitive way to understand this is that Nyström method utilizes a data-dependent basis, while random features use data-independent basis functions.

## D. Determine Inertial Frame Directions by Introducing Fourth Node

This appendix provides the formal geometric statement and proof for introducing a fourth node to disambiguate inertial-frame axis directions.

**Theorem D.1.** *For an inertial frame $F$, let $Q = [Q_0, Q_1, Q_2] \in \mathbb{R}^{3 \times 3}$ be the right-handed coordinate system formed by its principal axes. To uniquely determine the axis directions, we incorporate a fourth point whose coordinates in $Q$ have nonzero $x$- and $y$-components (equivalently, it does not lie on the $y$-$z$ plane or the $x$-$z$ plane), which removes the sign ambiguity of the canonical frame.*

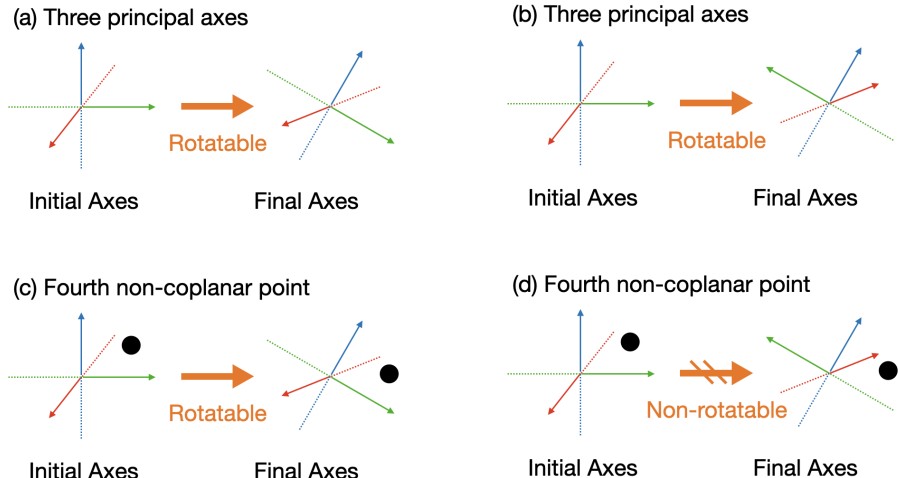

*Figure 7.* (a, b) show two potential rotational alignments between two coordinate systems (axes). (c, d) show that introducing a fourth point can remove the sign ambiguity and yield a unique alignment.

*Proof.* For three vectors, we can easily find a counterexample, as illustrated in Figure 7 (a, b). Figure 7 (a, b) describes two cases where we have the same initial frame, and we can rotate it to two different final frames with two rotation matrices, yet the right-handedness still matches. We can easily see that there are four options of rotation matrices in this case, and we cannot uniquely determine the final inertial frame.

More rigorously, let us assume that there exists a rotation matrix $R \in \mathrm{SO}(3)$ that transforms the initial coordinate system $Q_i = [Q_{i,0}, Q_{i,1}, Q_{i,2}] \in \mathbb{R}^{3 \times 3}$ to the final coordinate system $Q_f = [Q_{f,0}, Q_{f,1}, Q_{f,2}] \in \mathbb{R}^{3 \times 3}$ as:

$$Q_f = RQ_i. \tag{30}$$

First, we should change either zero or two directions for direction alignment. Then, without loss of generality, we can assume the two directions to be the last two axes. Thus, we can obtain a rotation matrix $R'$ such that $R'$ rotates $R$ around the axis $Q_{f,0} \in \mathbb{R}^3$ by 180 degrees. Using Rodrigues' rotation formula, this can be written as $R' = (2Q_{f,0}Q_{f,0}^T - I)R$. Thus, we can have:

$$R'Q_i = (2Q_{f,0}Q_{f,0}^T - I)Q_f = [Q_{f,0}, -Q_{f,1}, -Q_{f,2}]. \tag{31}$$

This is essentially saying that starting from one initial frame, we can have multiple matched final frames. Thus, using only three vectors cannot uniquely determine the direction matching. We provide two examples in Figure 7 (a, b).

For the four-vector case, we introduce an extra atom into the inertial frame system, whose coordinates in the canonical basis have nonzero $x$- and $y$-components. The problem becomes: starting from an initial frame and an extra point, can we find multiple rotation matrices such that the final frames have reflected directions? To be more rigorous, let us have the following formulation.

Let $\boldsymbol{v}_i$ denote the fourth point in the initial frame and let $\boldsymbol{v}_f$ denote its corresponding point in the final frame.

First, let us assume we have this rotation matrix:

$$[Q_{f,0}, Q_{f,1}, Q_{f,2}, \boldsymbol{v}_f] = R[Q_{i,0}, Q_{i,1}, Q_{i,2}, \boldsymbol{v}_i]. \tag{32}$$

As discussed above, we need to guarantee the right-handedness property, thus, without loss of generality, here we also assume the last two axes are reflected. The question turns to: does it exist another rotation matrix $R'$, such that:

$$[Q_{f,0}, -Q_{f,1}, -Q_{f,2}, \boldsymbol{v}_f] = R' [Q_{i,0}, Q_{i,1}, Q_{i,2}, \boldsymbol{v}_i]. \tag{33}$$

We now use contradiction. Since we still have the two axes rotated 180 degrees around the first axis $Q_{f,0}$, we have $R' = (2Q_{f,0}Q_{f,0}^T - I)R$. Then, from $\boldsymbol{v}_f = R\boldsymbol{v}_i$ and $\boldsymbol{v}_f = R'\boldsymbol{v}_i$, we have $(2Q_{f,0}Q_{f,0}^T - I)\boldsymbol{v}_f = \boldsymbol{v}_f$.

If we let $Q_{f,0} = [k_1, k_2, k_3]^T$ and $\boldsymbol{v}_f = [v_1, v_2, v_3]^T$, then we have

$$
\begin{aligned}
(2Q_{f,0}&Q_{f,0}^T - I)\boldsymbol{v}_f = \boldsymbol{v}_f \\
\begin{bmatrix} k_1k_1 & k_1k_2 & k_1k_3 \\ k_1k_2 & k_2k_2 & k_2k_3 \\ k_1k_3 & k_2k_3 & k_3k_3 \end{bmatrix} &\begin{bmatrix} v_1 \\ v_2 \\ v_3 \end{bmatrix} = \begin{bmatrix} v_1 \\ v_2 \\ v_3 \end{bmatrix} \\
\begin{bmatrix} k_1(k_1v_1 + k_2v_2 + k_3v_3) \\ k_2(k_1v_1 + k_2v_2 + k_3v_3) \\ k_3(k_1v_1 + k_2v_2 + k_3v_3) \end{bmatrix} &= \begin{bmatrix} v_1 \\ v_2 \\ v_3 \end{bmatrix} \\
(k_1v_1 + k_2v_2 + k_3v_3) \begin{bmatrix} k_1 \\ k_2 \\ k_3 \end{bmatrix} &= \begin{bmatrix} v_1 \\ v_2 \\ v_3 \end{bmatrix}.
\end{aligned}
\tag{34}
$$

After calculation, we can obtain that $Q_{f,0} = c\boldsymbol{v}_f$, where $c$ is a coefficient. However, by construction $\boldsymbol{v}_f$ does not lie on the same line as $Q_{f,0}$, thus, there does not exist such another rotation matrix $R' \neq R$ satisfying Equation (33). We also provide two examples in Figure 7 (c, d).

By contradiction, we can tell that there is only one unique rotation mapping from the initial inertial frame to the final inertial frame. $\square$

To sum up, three orthonormal basis vectors alone admit multiple valid sign assignments that preserve right-handedness, so the inertial-frame directions may be ambiguous. Introducing a fourth point whose canonical coordinates have nonzero $x$- and $y$-components removes this ambiguity by anchoring a unique orientation.

# E. Additional Analyses and Ablation Studies

In this section, we provide additional analyses, ablation studies, and robustness evaluations. Unless otherwise stated, the ablation experiments are performed on the QM9 unconditional generation setting, and we report the same four metrics as in the main paper: Valid, Valid&Unique, AtomSta, and MolSta. Note that the ablation runs were conducted under a slightly different experimental configuration from the main-table results; therefore, small numerical shifts in the "Ours" reference row relative to the headline tables are expected and do not affect the conclusions. For easy reference in the main text, we first report component-wise studies on inertial-frame robustness, GeoPE ablations (ManhattanPE and EuclideanPE), deterministic atom reordering, and Diffusion Loss for coordinates, and then summarize additional multi-seed stability and synthetic accessibility analyses.

## E.1. Robustness of the Canonical Inertial Frame

We perform two complementary analyses to assess the robustness of the canonical inertial frame: (i) stability under small geometric perturbations, and (ii) frequency of principal-moment degeneracy in realistic datasets.

**Stability under small perturbations.** We add i.i.d. Gaussian noise to atomic coordinates in QM9 and Drugs to quantify how often the "farthest atom" (used for axis-sign resolution) changes. Since quantum-derived coordinates are typically reported with precision around $10^{-3}$ Å, we consider perturbation magnitudes $\varepsilon \in [10^{-4}, 10^{-5}, 10^{-6}, 10^{-7}]$ Å, which are already larger than typical numerical noise and thus provide a conservative stress test of the sign-resolution step; even at the largest perturbation $\varepsilon = 10^{-4}$ Å, such changes remain rare. For each molecule and noise level, we measure the fraction of cases where the identity of the farthest atom changes relative to the unperturbed geometry.

As shown in Table 4, the sign-flip event becomes extremely rare at $\varepsilon = 10^{-5}$ Å (change ratio below $10^{-3}$ on QM9 and below $2 \times 10^{-5}$ on Drugs), and completely disappears at $\varepsilon = 10^{-7}$ Å. This indicates that the inertial-frame construction is highly stable under realistic coordinate noise.

*Table 4.* Farthest-atom change ratio under Gaussian coordinate perturbations.

| Dataset | $\varepsilon$ (Å) | Farthest-Atom Change Ratio |
|---|---|---|
| QM9 | $1 \times 10^{-4}$ | 0.00581 |
| | $1 \times 10^{-5}$ | 0.00078 |
| | $1 \times 10^{-6}$ | 0.00016 |
| | $1 \times 10^{-7}$ | 0.00000 |
| Drugs | $1 \times 10^{-4}$ | 0.00009900 |
| | $1 \times 10^{-5}$ | 0.00001980 |
| | $1 \times 10^{-6}$ | 0.00000375 |
| | $1 \times 10^{-7}$ | 0.00000000 |

**Principal-moment degeneracy.** We next quantify how often perfect symmetries (e.g., exact planarity or higher-order symmetry) cause principal-moment degeneracy, which in principle can make the inertial frame non-unique. We scan the full QM9 and Drugs datasets and count molecules with degenerate principal moments.

Table 5 shows that such cases are statistically negligible: only 9 molecules in QM9 (out of $\sim 130$K) and 1 molecule in Drugs exhibit exact degeneracy. These extremely rare symmetric molecules are simply excluded from training, which has no measurable impact on performance.

*Table 5.* Frequency of principal-moment degeneracy in QM9 and Drugs.

| Dataset | Number of Degenerate Molecules | Fraction |
|---|---|---|
| QM9 | 9 | 0.00007 |
| Drugs | 1 | 0.00000 |

Combining these analyses, the probability of principal-moment degeneracy is below $10^{-4}$ on QM9 and is effectively zero on Drugs. For sign flips, the farthest-atom change ratio drops below $10^{-3}$ at $\varepsilon = 10^{-5}$ Å and vanishes completely at

$\varepsilon = 10^{-7}$ Å. Empirically, we do not observe any training issues attributable to frame instability, supporting the practical robustness of our canonical inertial frame.

## E.2. Positional Encoding: GeoPE and Its Variants

We further ablate the proposed GeoPE by varying only the positional mechanism and keeping all other components fixed (inertial frame, hierarchical AR design, training setup, parameter count).

The compared variants are:

- **Ours**: full GeoPE (ManhattanPE + EuclideanPE).

- **No GeoPE**: 3D coordinates are encoded only as static features; the Transformer backbone uses no geometry-aware positional encoding.

- **ManhattanPE-only**: only the ManhattanPE component is used.

- **EuclideanPE-only**: only the EuclideanPE component is used.

The results in Table 6 highlight three key observations. First, removing GeoPE entirely causes a sharp drop in Valid and MolSta, indicating that a Transformer without geometry-aware positional structure cannot reliably reason about 3D molecular geometry. Second, both ManhattanPE-only and EuclideanPE-only models perform well, showing that each component provides a strong geometric inductive bias. Third, combining them into GeoPE yields the best overall performance, particularly in molecule-level stability. While the gains on QM9 appear modest, this is expected given that QM9 molecules are small and near-rigid; on more flexible datasets (e.g., Drugs, B3LYP-level systems), these geometric encodings are expected to play a larger role.

*Table 6.* Ablations on GeoPE on QM9.

| Model | Valid (%) | Valid&Unique (%) | AtomSta (%) | MolSta (%) |
|---|---|---|---|---|
| Ours (GeoPE) | **97.4** | **92.5** | **99.3** | **94.7** |
| No GeoPE | 8.7 | 3.8 | 20.2 | 0.0 |
| ManhattanPE-only | 97.1 | 92.5 | 99.2 | 94.3 |
| EuclideanPE-only | 97.3 | 92.5 | 99.2 | 94.2 |

These results support our claim that GeoPE is not merely a cosmetic design choice: it is the core mechanism that makes 3D molecular modeling feasible for an autoregressive Transformer.

## E.3. Diffusion Loss vs. Direct L2 Regression

To evaluate the coordinate prediction objective, we compare Diffusion Loss in our main model with a simple L2 regression loss on the coordinates. In the L2 variant, all other components—including the autoregressive architecture, inertial frame, and GeoPE—are kept identical.

*Table 7.* Comparison between diffusion loss and simple L2 coordinate regression on QM9.

| Model | Valid (%) | Valid&Unique (%) | AtomSta (%) | MolSta (%) |
|---|---|---|---|---|
| Diffusion Loss | **97.4** | **92.5** | **99.3** | **94.7** |
| L2 Loss | 24.7 | 4.4 | 76.2 | 14.2 |

As reported in Table 7, using an L2 loss causes a dramatic collapse in generation quality, showing that direct coordinate regression fails to model the distribution of 3D positions in an autoregressive setting. Diffusion Loss avoids this collapse and yields stable, valid structures, which is consistent with the insight reported in Li et al. (2024a). Therefore, we adopt Diffusion Loss in our framework.

### E.4. Effect of Canonical Atom Indexing

To evaluate the effect of canonicalizing atom indices, we compare the full model against a variant where the RDKit-based canonicalization step is removed while keeping all other components unchanged. In the non-canonical variant, atom indices are taken directly from the raw input ordering.

As shown in Table 8, removing canonicalization consistently degrades both validity and uniqueness, even though the drop is moderate in absolute terms. This confirms that enforcing a unique, RDKit-consistent atom ordering is beneficial for the autoregressive model, as it eliminates the $n!$ permutation ambiguity and provides a more stable training signal.

*Table 8.* Effect of RDKit-based canonicalization of atom indices on QM9.

| Model | Valid (%) | Valid&Unique (%) | AtomSta (%) | MolSta (%) |
|---|---|---|---|---|
| Ours (with canonicalization) | **97.4** | **92.5** | **99.3** | **94.7** |
| w/o canonicalization | 97.0 | 90.0 | 99.1 | 94.0 |

### E.5. Additional Multi-seed Stability Results

We provide additional multi-seed results to assess the stability of InertialAR. These runs follow the same method and evaluation protocol as the main paper, but were produced under the rebuttal experimental configuration; therefore, small numerical shifts relative to the headline tables may occur. QM9 uses 5 random seeds, while GEOM-Drugs, B3LYP-1M, and class-conditional QM9 use 3 random seeds due to computational cost. Across all settings, the conclusions remain stable and are not driven by a favorable single run.

*Table 9.* Additional multi-seed mean±std for the unconditional generation results of InertialAR.

| Dataset | Valid (%) | Valid&Uni (%) | AtomSta (%) | MolSta (%) |
|---|---|---|---|---|
| QM9 | 97.37±0.14 | 92.29±0.20 | 99.31±0.05 | 94.51±0.26 |
| GEOM-Drugs | 96.74±0.27 | – | 86.97±0.26 | – |
| B3LYP-1M | 99.07±0.21 | 98.55±0.25 | 84.69±0.12 | 23.96±0.40 |

*Table 10.* Additional multi-seed mean±std for the class-conditional generation results of InertialAR on QM9.

| Class | Hit Rate (%) | Valid (%) | Valid&Uni (%) | AtomSta (%) | MolSta (%) |
|---|---|---|---|---|---|
| 7 | 90.34±0.62 | 98.76±0.31 | 92.18±0.54 | 99.62±0.14 | 96.89±0.48 |
| 28 | 89.15±0.58 | 99.72±0.23 | 90.03±0.61 | 99.85±0.11 | 98.74±0.37 |
| 3 | 85.08±0.71 | 99.81±0.17 | 86.24±0.58 | 99.87±0.09 | 98.96±0.42 |
| 16 | 67.83±0.84 | 91.62±0.53 | 78.55±0.67 | 96.84±0.26 | 80.31±0.59 |
| 23 | 81.12±0.65 | 99.48±0.28 | 82.03±0.56 | 99.86±0.12 | 98.67±0.44 |

### E.6. Synthetic Accessibility under Conditional Generation

*Table 11.* Synthetic accessibility (SA) scores on QM9.

| Setting | Uncond. | Class 3 | Class 7 | Class 16 | Class 23 | Class 28 |
|---|---|---|---|---|---|---|
| **SA Score ($\downarrow$)** | 4.28 | 4.13 | 4.51 | 3.42 | 4.78 | 4.42 |

To complement validity and stability metrics, we report the RDKit synthetic accessibility (SA) score on valid generated molecules for unconditional and class-conditional generation on QM9. Lower is better. We emphasize that SA is only a 2D graph-level proxy and does not replace retrosynthetic or experimental validation.

The class-conditional scores remain broadly comparable to the unconditional baseline overall, suggesting no severe degradation in this proxy under CFG-driven control.

