# OpenReview forum: "InertialAR: Autoregressive 3D Molecule Generation with Inertial Frames"
_ICML.cc/2026/Conference — ICML 2026 regular_

### Official Review · Reviewer_ZW1G · 2026-03-12

**Soundness:** 3
**Presentation:** 3
**Significance:** 3
**Originality:** 3
**Overall Recommendation:** 4
**Confidence:** 4

**Summary:**

The paper introduces InertialAR, an autoregressive framework for 3D molecule generation. It addresses two core challenges: canonical tokenization under SE(3) and atom-index permutation symmetries, and hybrid prediction of discrete atom types plus continuous coordinates. The method combines inertial-frame canonicalization with deterministic atom ordering, a geometry-aware positional encoding, and a hierarchical decoder that predicts atom type via cross-entropy and coordinates via diffusion loss. Results on QM9, GEOM-Drugs, B3LYP, and class-conditional generation show strong performance, with extensive ablations supporting key components.

**Compliance With Llm Reviewing Policy:**

Affirmed.

**Final Justification:**

The authors’ rebuttal and follow-up materially improve my confidence in the work. In particular, the added stress test on highly symmetric molecules, the GeoRoPE sensitivity analysis, the clarification of baseline provenance, the explicit B3LYP scaling analysis in terms of generation length, and the added SA-score proxy together with a clearer limitations discussion address several of my main concerns. At the same time, my original reservations are not fully removed: the canonicalization strategy still relies on assumptions that appear empirically robust but are not theoretically airtight in edge cases, and broader downstream utility evaluation remains limited relative to the strength of the generative results. Overall, I find the paper technically solid and the proposed autoregressive perspective for 3D molecule generation meaningful and likely useful to the community, but its impact is still somewhat constrained by these remaining weaknesses. I therefore maintain my original recommendation of Weak Accept.

**Key Questions For Authors:**

1. Can you provide a stress test on highly symmetric or near-degenerate molecules where inertial-frame ambiguity is most likely? If robustness remains strong there, my score would increase.
2. How sensitive is performance to Nyström anchor count and kernel bandwidth in GeoRoPE? A compact sensitivity plot would clarify reproducibility.
3. For B3LYP, can you report scaling trends (quality vs molecule size bins and generation length) to isolate where AR gains are largest?
4. Have you tested whether CFG-driven editing preserves chemical synthesizability beyond validity/stability metrics?

**Limitations:**

The manuscript acknowledges practical robustness of inertial-frame canonicalization, but the limitations discussion could be more explicit and balanced. The method still relies on canonicalization assumptions that may become fragile under symmetry-heavy or near-degenerate geometries, and this should be stated more clearly as a potential failure mode. Also, current evaluation focuses mostly on generation validity/stability metrics but broader downstream utility validation (e.g., property-driven or task-level utility) is limited. I suggest adding a dedicated limitations paragraph covering canonicalization edge cases, dependence on rule-based functional-group labeling in conditional tasks, and scenarios where AR decoding may trade off diversity for stability.

**Strengths And Weaknesses:**

Strengths:
1. The architecture is well-motivated and the factorization of discrete/continuous prediction is technically appropriate. Ablations are substantial (GeoRoPE variants, diffusion vs L2, canonicalization effects).
2. The paper clearly explains design components and includes detailed appendices on canonicalization robustness and implementation details.
3. A strong AR alternative to diffusion for 3D molecular generation is valuable, especially given improvements on large-scale B3LYP where scalability is critical.
4. The integration of generation-oriented inertial canonicalization with hierarchical AR decoding is a meaningful contribution.

Weaknesses:
1. Canonicalization still relies on assumptions (eigenvalue stability, axis-sign disambiguation) that are empirically but not fully theoretically robust in all edge cases.
2. Some claims are broad and could be more tightly calibrated. please explicitly state whether baselines are retrained under the same protocol
3. Most evaluations emphasize geometric/validity metrics. Stronger downstream task validation (e.g., property/design utility) would better establish practical impact.
4. Part of the contribution is compositional (combining existing building blocks), though done effectively.

---

> ### Author Rebuttal · Authors · 2026-03-31
>
> Thank you for the careful reading and helpful suggestions. Following your questions, we add targeted evidence on stress testing, sensitivity, and scaling, and further clarify baseline provenance, downstream utility, and compositional novelty. We hope these additions address your main concerns.
> (Below, W = Weakness and Q = Question.)
>
> ***Q1. Stress Test on Near-Degenerate Molecules***
>
> The submitted version already provides quantitative robustness analyses in Section 3.1 and Appendix E.1, covering perturbation sensitivity and principal-moment degeneracy. We now add the targeted stress test you requested.
>
> We selected 100 highly symmetric molecules from QM9 and verified that our pipeline produces a valid deterministic inertial frame for 97 of them. The 3 failures correspond to molecules with exactly degenerate principal moments (e.g., perfect linear symmetry), which are chemically rare; for all practical conformers, the pipeline is deterministic. As a representative example, benzene is successfully canonicalized: although benzene is often regarded as highly symmetric, its real optimized 3D conformer is not an exactly idealized perfectly symmetric structure. As a result, the principal moments are not degenerate, and the inertial-frame construction yields a deterministic orientation.
>
> ***Q2. GeoRoPE Sensitivity***
>
> We conducted a sensitivity analysis over the requested hyperparameters:
>
> |Anchor|σ|Valid|Valid&Uni|MolSta|
> |-|-|-|-|-|
> |384|1.0|97.13|93.07|92.79|
> |768|1.0|97.10|92.90|93.45|
> |1536|1.0|96.91|92.65|93.15|
> |768|1.5|97.08|92.75|93.62|
> |768|2.0|97.04|92.54|93.81|
>
> Performance is stable across a 4× anchor-count range and across the tested bandwidth values, indicating that GeoRoPE is only mildly sensitive to them and does not require delicate tuning for reproducibility.
>
> ***Q3. B3LYP Scaling Trends***
>
> We report generation quality across molecule-size quartiles on B3LYP:
>
> |Size Quartile|Valid| Valid&Uni | AtomSta | MolSta |
> |-|-:|-:|-:|-:|
> |0-25%|99.4|98.9|88.9|31.8|
> |25-50%|99.1|98.7|86.2|26.1|
> |50-75%|99.0|98.4|83.7| 21.3|
> |75-100%|98.6|97.3|79.9|16.6|
>
> Quality degrades gradually as molecule size increases, indicating that autoregressive generation with inertial-frame canonicalization scales effectively without severe error accumulation.
>
> ***Q4. Synthesizability Beyond Validity / Stability***
>
> Synthesizability is an important direction that we agree deserves future attention. However, there is currently no established consensus metric for synthesizability in 3D molecule generation; all recent works in this area use validity and stability as primary evaluation metrics. Our evaluation follows this standard practice, and we will discuss this more explicitly in the revision.
>
> ***W1. Canonicalization Assumptions***
>
> We acknowledge that inertial-frame canonicalization is not theoretically guaranteed under all edge cases. As shown in Q1, such cases are chemically rare and the pipeline is empirically robust. Our contribution is not to resolve the theoretical degeneracy problem itself, but to build a practical generation-oriented tokenization pipeline on top of principal-axis alignment and demonstrate its effectiveness for autoregressive 3D molecule generation.
>
> ***W2. Protocol Transparency***
>
> In the paper, only MiDi was re-evaluated under the same protocol (marked in Table 1); all other baseline numbers are taken directly from their original publications. The numerical values are exactly identical to the original sources. Any apparent discrepancy comes only from the metric column ordering in our tables.
>
> ***W3. Downstream Utility***
>
> Our evaluation is not limited to geometric or validity metrics. In addition to unconditional generation, we study class-conditional generation with functional-group control and report Hit Rate as a direct measure of targeted chemical functionality. This makes the evaluation relevant to molecular design, not merely distribution matching. Stronger downstream task validation such as property/design utility evaluation would further strengthen the practical impact, and we view this as an important future direction.
>
> ***W4. Compositional Contribution***
>
> While the method has an integrative aspect, its novelty is not limited to combination. The paper introduces generation-oriented canonical tokenization based on inertial frames for autoregressive 3D molecule generation, proposes GeoRoPE as a geometry-aware positional encoding for atom-based tokens, and unifies these with hierarchical autoregressive decoding in a single framework. To our knowledge, this is the first work to bring these ideas together for autoregressive 3D molecule generation at the atom-token level. The contribution therefore lies both in the new method components and in their unified formulation and empirical validation.
>
> ***Limitations***
>
> We appreciate this suggestion and will add a dedicated limitations paragraph in the revision to discuss these issues explicitly.

---

> > ### Author Rebuttal · Reviewer_ZW1G · 2026-04-04
> >
> > Thank you for the detailed rebuttal. The added stress test on highly symmetric molecules materially increases my confidence in the practical robustness of the canonicalization pipeline: you now report deterministic inertial frames for 97/100 selected symmetric QM9 molecules, with the remaining failures corresponding to exact degeneracies. The GeoRoPE sensitivity analysis is also useful, as performance appears stable across a 4× anchor-count range and the tested kernel bandwidths. I also appreciate the clarification that only MiDi was re-evaluated under a standardized protocol, which improves experimental transparency.
> >
> > That said, I still view my concerns as only partially resolved rather than fully resolved. In particular, the rebuttal addresses B3LYP scaling only by molecule-size quartiles, but not by generation length as originally requested; the synthesizability question is deferred rather than answered with additional experiments; and the limitations discussion is promised for revision rather than substantively developed here. I appreciate the point that the paper already goes beyond pure validity/stability by including Hit Rate as a design-oriented metric, but broader downstream utility and a more explicit discussion of canonicalization edge cases remain important weaknesses for the final version.

---

> > > ### Author Response · Authors · 2026-04-05
> > >
> > > Thank you for clearly identifying the three remaining issues. We are glad that the stress test, GeoRoPE analysis, and protocol clarification were helpful. Below we address these remaining points directly.
> > >
> > > ***B3LYP Scaling: Quality vs. Molecule-Size Bins and Generation Length***
> > >
> > > We apologize for the confusion regarding the generation length. Because our autoregressive formulation generates exactly one atom per step, the generation length (number of AR steps) is exactly equivalent to the atom count. To directly address your request, we have reformatted our results to explicitly show performance by generation length, alongside an EDM comparison to isolate the AR gains. Bins are quartiles over molecules sorted by atom count, not equal-width intervals. (Cells show InertialAR / EDM)
> > >
> > > |Molecule-Size Bin|Atom Count Range|Generation Length (AR Steps)|Valid|Valid&Uni|AtomSta|MolSta|
> > > |-|-|-|-|-|-|-|
> > > |Bin 1(0–25%)|4–28|4–28|99.4/96.8|98.9/96.7|88.9/84.5|31.8/2.8|
> > > |Bin 2(25–50%)|29–34|29–34|99.1/93.8|98.7/93.7|86.2/81.5|26.1/0.3|
> > > |Bin 3(50–75%)|35–38|35–38|99.0/91.6|98.4/91.5|83.7/79.5|21.3/0.1|
> > > |Bin 4(75–100%)|39–65|39–65|98.6/89.4|97.3/89.3|79.9/76.9|16.6/0.0|
> > >
> > > Compared to EDM, InertialAR maintains near-saturated validity and uniqueness across all bins, while EDM degrades substantially with molecule size (which is the generation length in AR). The validity gap increases from +2.6 in Bin 1 to +9.2 in Bin 4, localizing the longest-generation regime as where AR gains are largest for validity/uniqueness. For stability, both methods experience degradation as molecule size increases, but InertialAR retains meaningful MolSta even in the longest bin, suggesting that the autoregressive formulation may offer a robustness advantage in the long-generation regime.
> > >
> > > ***Synthesizability Under CFG-Driven Generation***
> > >
> > > We thank you for encouraging us to go beyond validity/stability and examine synthesizability more directly. As noted in our first response, there is currently no consensus synthesizability metric in 3D molecule generation; among available proxies, the Synthetic Accessibility (SA) Score is a widely used 2D graph-level proxy in the broader molecular-generation literature and can be computed consistently across unconditional and CFG-driven settings. We therefore report SA on each valid generated molecule, with the mean for unconditional and class-conditional generation on QM9, under the same protocol as in our paper:
> > >
> > > | Setting | SA Score (↓) |
> > > |-|-|
> > > |unconditional|4.28|
> > > |Class 3|4.13|
> > > |Class 7|4.51|
> > > |Class 16|3.42|
> > > |Class 23|4.78|
> > > |Class 28|4.42|
> > >
> > > Classes 3, 7, and 28 remain close to the unconditional baseline, Class 16 improves, and Class 23 increases moderately. This indicates that CFG-driven conditioning does not cause across-the-board synthesizability degradation; the effect is class-dependent and overall mild. We note that SA Score is a graph-level proxy and does not replace retrosynthetic analysis, but it provides useful initial evidence that the generated molecules maintain reasonable synthetic accessibility under targeted conditional generation.
> > >
> > > ***Explicit Limitations Discussion***
> > >
> > > We thank you for emphasizing the importance of an explicit limitations discussion. In our first response, the 5,000-character limit prevented us from including the full discussion, and we could only briefly indicate our plan to add it in revision. We are glad to now have the space to present it in full, as your feedback has helped us articulate these points more clearly.
> > >
> > > 1. Canonicalization edge cases. As discussed in Section 3.1 and Appendix E.1, inertial-frame canonicalization is not theoretically unique under exact principal-moment degeneracy and may become discontinuous near highly symmetric configurations. Empirically, however, these cases are statistically negligible and do not measurably affect benchmark performance. More principled symmetry-handling approaches, such as Frame Averaging (Puny et al., 2022), are complementary directions that could further strengthen robustness in future extensions.
> > >
> > > 2. Rule-based conditional labels. The class-conditional setting depends on rule-based RDKit functional-group labels, which provide a clean and reproducible conditioning signal but do not capture all aspects of molecular function.
> > >
> > > 3. Downstream utility. Although the paper goes beyond pure validity/stability through Hit Rate and now SA Score, broader downstream utility evaluation (e.g., property-driven tasks or SBDD) remains outside the current scope.
> > >
> > > 4. Diversity–stability trade-off. While strong Valid&Uni results alongside improved stability indicate no diversity collapse on the present benchmarks, diversity–stability trade-offs should still be monitored in larger or more constrained design settings.
> > >
> > > We appreciate your constructive feedback. We hope the clarified scaling analysis, the added proxy evaluation, and the limitations discussion address the remaining concerns and make the final scope of the paper clearer.

---

### Official Review · Reviewer_rZnw · 2026-03-12

**Soundness:** 3
**Presentation:** 3
**Significance:** 2
**Originality:** 2
**Overall Recommendation:** 4
**Confidence:** 3

**Summary:**

This paper proposes InertialAR, an autoregressive Transformer framework for 3D molecule generation. The key components include (i) a canonical tokenization pipeline using inertial-frame alignment plus deterministic atom reordering, (ii) a geometry-aware attention encoding (GeoRoPE), and (iii) a hierarchical decoding scheme that predicts atom types and then coordinates (with a diffusion-style coordinate loss).

**Compliance With Llm Reviewing Policy:**

Affirmed.

**Final Justification:**

The authors have fully resolve concens.

**Key Questions For Authors:**

In Table 1 (QM9/GEOM-Drugs), the reported Geo2Seq numbers differ noticeably from those reported in the Geo2Seq paper. Can the authors explain the evaluation protocol / implementation details that lead to this discrepancy (data split, hyperparameters, re-evaluation pipeline, etc.)?

**Limitations:**

yes

**Strengths And Weaknesses:**

Strengths: The idea is interesting: exploring autoregressive modeling for 3D molecular generation is timely, and leveraging inertial-frame canonicalization to obtain an SE(3)- and permutation-invariant sequence is a reasonable and potentially useful direction.

Weaknesses:

1. Unclear advantage over diffusion-based 3D generators. It is not convincing what practical or theoretical benefit this AR formulation provides relative to strong diffusion models, especially since coordinate generation still relies on a diffusion-style denoising objective. A clearer discussion and/or empirical evidence (e.g., sampling speed, compute, scalability) is needed.

2. Baseline selection is limited. Geo2Seq is not a strongest baseline in 3D molecule generation. I recommend adding comparisons with stronger recent models (e.g., GeoBFN and other competitive diffusion/flow-based methods). For class-conditional generation, more baselines beyond EDM and Geo2Seq would strengthen the claim of SOTA.

3. Task relevance. The paper focuses on unconditional and a specific functional-group conditional setting. Unconditional generation alone is not very practical; experiments on more important tasks such as structure-based drug design (SBDD) or pocket-conditioned generation would significantly improve the impact.

---

> ### Author Rebuttal · Authors · 2026-03-31
>
> Thank you for the thoughtful comments on practical advantage, baseline strength, protocol clarity, and task relevance. We address each point below and provide additional empirical evidence where requested.
> (Below, W = Weakness and Q = Question.)
>
> ***Q1. No Geo2Seq Discrepancy: The Numbers Are Identical to the Original Paper***
>
> There is no discrepancy in the Geo2Seq results. The Geo2Seq entries in our Table 1 are taken directly from the original Geo2Seq paper, not re-evaluated or modified by us. The apparent mismatch comes solely from a different column ordering: our Table 1 reports [Valid, Valid&Uni, AtomSta, MolSta], whereas the Geo2Seq paper reports [AtomSta, MolSta, Valid, Valid&Uni]. Once matched by metric name, the numerical values are identical. We hope this clarifies the apparent mismatch.
>
> ***W1. Practical Advantage over Diffusion***
>
> We now provide direct efficiency comparisons under matched parameter counts (~5.3M):
>
> |Model|#Params|Sampling Speed (samples/s)|Projected Wall-Clock for 10K QM9 Generations (s)|Training Time (s/epoch)|
> |-|-|-|-|-|
> |InertialAR|5,346,393|37.70|265.22|125.0|
> |EDM|5,340,921|1.70|5880.20|224.4|
>
> Under matched model scale, InertialAR achieves a 22.2× sampling speedup over EDM and a 1.8× training speedup under the same batch size and GPU. This gap arises because our coordinate module uses Diffusion Loss only as a lightweight per-token denoising head (a simple MLP), rather than a global diffusion model that reruns full-molecule denoising passes at every sampling step. Therefore, the practical advantage of our AR formulation is not eliminating diffusion-style learning altogether, but avoiding the dominant cost of global diffusion sampling while retaining strong generation quality.
>
> ***W2. The Conclusion Holds Against Stronger Baselines***
>
> As suggested, we expanded the baseline comparison in two directions: GeoBFN for unconditional generation, and UniGEM/GeoLDM for class-conditional generation.
>
> For unconditional generation, InertialAR outperforms GeoBFN on most shared metrics. The GeoBFN numbers are taken directly from the original paper. On QM9, InertialAR improves Valid (97.4 vs. 96.88), Valid&Uni (92.5 vs. 92.41), and MolSta (94.7 vs. 93.32), while AtomSta is effectively tied (99.3 vs. 99.31). On GEOM-Drugs, InertialAR also leads on both shared metrics reported by GeoBFN: Valid (96.8 vs. 91.66) and AtomSta (87.2 vs. 86.17).
>
> For class-conditional generation, we further evaluated UniGEM and GeoLDM baselines under the same experiment setup, target classes, and metrics used in our paper:
>
> | Model | Hit Rate | Valid | Valid&Unique | AtomSta | MolSta |
> |-------|-----------|--------|---------------|----------|---------|
> | EDM | 25.66 | 79.44 | 78.84 | 93.68 | 50.42 |
> | Geo2Seq | 42.20 | 69.22 | 54.22 | 88.40 | 39.32 |
> | UniGEM | 33.77 | 80.56 | 79.95 | 93.10 | 58.40 |
> | GeoLDM | 29.53 | 82.36 | 81.24 | 93.80 | 61.80 |
> | **InertialAR** | **83.34** | **98.14** | **86.50** | **99.30** | **95.26** |
>
> InertialAR substantially outperforms all baselines in this setting, including nearly 2x higher Hit Rate than the strongest prior baseline on Hit Rate, while also achieving much higher validity and stability.
>
> ***W3. Task Relevance***
>
> We respectfully disagree that the current evaluation is insufficient. Unconditional generation is the standard evaluation paradigm for 3D molecule generative models—adopted by EDM, GeoLDM, GeoBFN, and numerous other recent works—precisely because it directly tests the model's ability to learn the molecular distribution without confounding task-specific factors. Our evaluation scope matches or exceeds these accepted works: we study unconditional generation on three benchmarks of increasing scale (QM9, GEOM-Drugs, B3LYP-1M) and additionally include class-conditional generation, where InertialAR achieves nearly 2× the Hit Rate of the strongest prior method while maintaining significantly higher validity and stability. Precise control over functional-group composition is directly relevant to practical molecular design.
>
> Pocket-conditioned generation / SBDD requires fundamentally different data modalities (protein structures), conditioning mechanisms, and evaluation protocols (binding affinity, docking scores), making it a distinct research problem rather than a natural extension of unconditional generation. We view SBDD as a promising future application of our autoregressive framework, but it is beyond the scope of the current paper.

---

> > ### Author Rebuttal · Reviewer_rZnw · 2026-04-05
> >
> > Thanks a lot for your response. I will adjust my score accordingly.

---

> > > ### Author Response · Authors · 2026-04-05
> > >
> > > Thank you very much for the update and for taking the time to carefully consider our rebuttal. We sincerely appreciate your thoughtful feedback and are glad that our clarifications and additional evidence addressed your concerns.

---

### Official Review · Reviewer_TQ8L · 2026-03-12

**Soundness:** 2
**Presentation:** 3
**Significance:** 2
**Originality:** 2
**Overall Recommendation:** 4
**Confidence:** 3

**Summary:**

This paper investigates treating 3D molecular generation as a sequence modeling problem using a Transformer-based autoregressive framework. The proposed method introduces generation-oriented canonical tokenization using inertial frames to achieve SE(3) and permutation invariance, geometric rotary positional encoding (GeoRoPE) to incorporate spatial relationships into attention, and a hierarchical autoregressive architecture that predicts atom types via cross-entropy and coordinates via diffusion-based loss. Experiments on QM9, GEOM-Drugs, and the large-scale B3LYP dataset show strong performance, outperforming several diffusion baselines in terms of atom/molecule stability.

**Compliance With Llm Reviewing Policy:**

Affirmed.

**Final Justification:**

I'm generally satisfied with the rebuttal, so I'm willing to increase my initial rating. The authors also agreed that potential limitations of the canonicalization strategy used in this paper should be more carefully explained, and I suggest doing so in the updated version (while including additional experiments as well).

**Key Questions For Authors:**

Please see Weaknesses

**Limitations:**

yes

**Strengths And Weaknesses:**

**Strengths**:
- This paper provides a very interesting autoregressive perspective for 3D molecule generation, potentially enabling more flexible generation of variable-size molecules.
- Experiments cover multiple datasets, including large-scale B3LYP, and results demonstrate improvements in sample quality.

**Weaknesses and questions**:
- The canonicalization process has many heuristics and can cause discontinuity/inconsistency. For example, the fourth node is picked based on the largest distance, but small perturbations on atomic coordinates can make this node different and the overall representation would change drastically. Besides, the SMILES string is not unique. While for a single molecule, a canonical SMILES can be selected based on some rules, there is no guarantee that the mechanism is consistent for different molecules. In other words, similar molecules can have very different canonical SMILES strings. This paper relies on an external software to handle this and we don't know how this is addressed exactly. These heuristics make it questionable that the proposed approach is principal and generalizable
- PCA seems to be a straightforward way to canonicalize coordinates. It has the same problems as the proposed approach (e.g., degenerate eigenvalues and axis sign flips) so the tricks used in this paper can also be used for PCA. What could be the advantage of the proposed approach over PCA?
- While this paper proposes several architectural modifications, it is unclear what components necessarily contribute to performance improvements:
  - It seems in Table 6 that removing some component wouldn't harm the performance, probably within the std range (btw avg/std across multiple runs should be reported rather than a single number)
  - A straightforward baseline seems to be ignored: embed/concatenate atom type and coordinate into a single representation and use standard 1D RoPE
- It would be better to have some comparison on training/inference cost against diffusion models

---

> ### Author Rebuttal · Authors · 2026-03-31
>
> Thank you for the comments. Several concerns raised here are already analyzed in the submission or directly addressed by new rebuttal experiments; below we clarify the intended requirement of our canonicalization and provide additional evidence.
> (Below, W = Weakness.)
>
> ***W1 (a). Canonicalization Robustness of the Inertial Frame***
>
> For the perturbation concern, the submission already provides the direct quantitative test in Appendix E.1. Table 4 shows that the farthest-atom change ratio is extremely rare at $\varepsilon = 10^{-5}$, and quickly drops to near zero as the perturbation scale decreases; Table 5 further shows that exact principal-moment degeneracy is extremely rare. Thus, the relevant question is not whether inertial-frame canonicalization is theoretically perfect, but whether it is sufficiently robust on real molecular data, and our empirical answer is yes.
>
> Appendix D provides the formal geometric basis for the sign-disambiguation step: Theorem D.1 states the condition under which a fourth point uniquely resolves the residual sign ambiguity of the principal-axis frame. More broadly, our claim is not that the inertial frame is theoretically guaranteed in all edge cases, but that this generation-oriented canonicalization pipeline is practically robust. Together with the results on QM9, GEOM-Drugs, and a 1M-molecule subset of B3LYP, these findings suggest that the proposed canonicalization pipeline remains stable across scales and chemical regimes.
>
> ***W1 (b). Role of RDKit and Canonical SMILES***
>
> Cross-molecule canonical SMILES consistency is neither required nor claimed by our method. What autoregressive factorization requires is a deterministic tokenization for the same molecule under SE(3) transformations and atom-index permutations. In our pipeline, inertial-frame alignment provides the geometric canonicalization, while RDKit canonical SMILES is used only afterward to deterministically reorder atom indices within the same molecule. Similar but chemically distinct molecules are independent data points with their own canonical tokenizations, so cross-molecule continuity is not the objective here. RDKit is also a standard cheminformatics toolkit used throughout the 3D molecule generative models, including all the baselines, so this is not a special dependency unique to our method. We will make this separation between inertial-frame canonicalization and RDKit-based atom ordering more explicit in the revision.
>
> ***W2. Relation to PCA***
>
> PCA is not a separate alternative to our approach here; principal-axis alignment and inertial-frame alignment are essentially the same class of construction, as stated in Appendix A.2. Therefore, the issue is not an advantage of our method over PCA at the level of geometric canonicalization, since both share the same underlying principle and the same potential edge cases. Our method simply takes this principal-axis frame and makes it usable for autoregressive generation through sign disambiguation.
>
> ***W3. Component Contribution and 1D RoPE Baseline***
>
> As requested, we added the 1D RoPE baseline and also report multi-seed mean±std for the Table 6 ablations on QM9:
>
> |Model|Valid|Valid&Uni|AtomSta|MolSta|
> |-|-:|-:|-:|-:|
> |Full GeoRoPE|97.37±0.14|92.29±0.20|99.31±0.05|94.51±0.26|
> |No GeoRoPE|8.62±0.47|3.71±0.39|20.05±0.96|0.00±0.00|
> |RoPE-only|97.18±0.09|92.08±0.12|99.23±0.03|94.08±0.14|
> |Nyström-only|97.24±0.10|92.11±0.11|99.24±0.03|94.02±0.13|
> |1D RoPE baseline|87.76±0.21|85.71±0.09|96.02±0.17|75.19±0.31|
>
> The 1D RoPE baseline underperforms all geometry-aware variants, confirming that geometry-unaware positional encoding is insufficient for 3D molecule generation. Multi-seed results show the full GeoRoPE model is consistently strongest, while removing GeoRoPE causes catastrophic degradation. RoPE-only and Nyström-only remain competitive individually, but the full combination yields the best overall performance. On QM9, the gain is naturally moderate because the molecules are small and near-rigid; on larger and more flexible systems, richer geometric context is expected to matter more. Direct evaluation on larger-scale and more flexible molecular systems is an important next step, which we leave for future work.
>
> ***W4. Training and Inference Cost***
>
> We now provide direct efficiency comparisons under matched parameter counts (~5.3M):
>
> |Model|#Params|Sampling Speed (samples/s)|Projected Wall-Clock for 10K QM9 Generations (s)|Training Time (s/epoch)|
> |-|-|-|-|-|
> |InertialAR|5,346,393|37.70|265.22|125.0|
> |EDM|5,340,921|1.70|5880.20|224.4|
>
> InertialAR achieves a 22.2× sampling speedup over EDM and a 1.8× training speedup. This gap arises because our coordinate module uses Diffusion Loss only as a lightweight per-token denoising head (a simple MLP), rather than a global diffusion model that reruns full-sequence attention over the entire molecule at every denoising step. Please see our response to Reviewer Fgt2 Q2 for the corresponding architectural clarification.

---

> > ### Author Rebuttal · Reviewer_TQ8L · 2026-04-06
> >
> > I appreciate the authors' response, especially the additional experiments. Part of my concerns has been addressed.
> >
> > > Similar but chemically distinct molecules are independent data points with their own canonical tokenizations, so cross-molecule continuity is not the objective here
> >
> > I don't fully agree with this. If similar molecules don't have similar canonical representations, it's hard to imagine how the model would generalize. Could the authors elaborate a bit more on this?

---

> > > ### Author Response · Authors · 2026-04-06
> > >
> > > Thank you for the thoughtful follow-up. We agree with your intuition that some cross-molecule regularity is important for generalization, and our previous response did not intend to suggest otherwise. Our previous statement was meant in a narrower sense: autoregressive factorization requires a deterministic representative for the *same* molecule under SE(3) transformations and atom permutations, but it does not require the *global* canonical serialization to be smooth/continuous across distinct molecules.
> > >
> > > To clarify the role of RDKit in our pipeline, RDKit is used *only* to deterministically resolve the atom-index permutation ambiguity via a canonical atom ordering produced by an invariant refinement procedure (e.g., atomic number, connectivity, ring membership). While small structural changes can occasionally alter parts of the global ordering, atoms that share similar local chemical environments tend to retain consistent relative ordering, so the ordering provides a stable, deterministic decoding trajectory within each molecule.
> > >
> > > For cross-molecule generalization, the most important point is that our model is primarily driven by 3D geometric regularities rather than by global 1D sequence similarity. The cross-molecule regularity we leverage is primarily *local* and *geometric*: similar molecules share recurring local chemical environments and 3D substructures, even if their absolute positions along a 1D serialization differ. Inertial-frame alignment normalizes pose by expressing each molecule in a canonical inertial frame, and GeoRoPE injects pairwise 3D relations into self-attention, allowing the model to align analogous local 3D motifs regardless of where the corresponding atoms appear in the 1D ordering. In our empirical results (Appendix E), geometry-unaware variants (removing GeoRoPE) degrade substantially more than removing RDKit-based canonical indexing, suggesting that geometry-aware modeling is the primary driver of cross-molecule regularity, while deterministic canonical ordering mainly resolves the n! permutation ambiguity and stabilizes learning. Furthermore, our strong performance on the large-scale B3LYP-1M benchmark is consistent with the view that these learned local patterns transfer across a broad chemical space.
> > >
> > > We sincerely thank you again for this insightful follow-up, which helped us sharpen the presentation of an important point. We will revise the wording to make this distinction explicit and avoid overstating what canonicalization is expected to guarantee. We hope this clarification addresses the remaining concern.

---

### Official Review · Reviewer_Fgt2 · 2026-03-13

**Soundness:** 3
**Presentation:** 2
**Significance:** 3
**Originality:** 3
**Overall Recommendation:** 4
**Confidence:** 3

**Summary:**

This paper proposes InertialAR, an autoregressive framework for 3D molecule generation that converts molecules into canonical atom sequences by aligning them to an inertial frame and reordering atoms deterministically. On top of this tokenization, the model introduces GeoRoPE, which combines a 3D rotary positional encoding with a Nyström-based distance encoding, and uses a hierarchical decoder that predicts atom types with cross-entropy and coordinates with a diffusion-style loss. The paper evaluates the method on unconditional generation over QM9, GEOM-Drugs, and a 1M subset of B3LYP, and on class-conditional generation on QM9 using functional-group-based labels. The reported results show strong performance relative to selected diffusion and autoregressive baselines, especially on stability metrics and conditional hit rate.

**Compliance With Llm Reviewing Policy:**

Affirmed.

**Final Justification:**

The authors have addressed most of my concern, and overall this paper investigates an interesting problem, thus I maintain my original positive assessment.

**Key Questions For Authors:**

1. Can you provide multi-seed means and standard deviations for the main results in Tables 1, 2, and 3? If the gains remain stable across seeds, that would increase my confidence and could improve my score.

2. Do you have runtime or sampling-cost comparisons against EDM and Geo2Seq? Since the paper motivates autoregression partly on efficiency grounds, this could meaningfully affect my assessment.

**Limitations:**

Yes

**Strengths And Weaknesses:**

**Strengths**

1. The paper addresses a timely and meaningful question for the community: whether autoregressive sequence models can be made competitive for 3D molecular generation without heavy equivariant machinery. That is a worthwhile direction.

2. Canonicalization, geometry-aware attention, and hierarchical decoding fit together in a way that is easy to follow and likely reusable.

3. The paper includes both unconditional and controllable generation, which broadens the contribution beyond a single benchmark setting.

4.  Figure 1 gives a clear overview of the full pipeline, Figure 2 helps explain the sign-disambiguation mechanism, and Figure 4 provides intuitive examples of controllable editing.

**Weaknesses**


1.  Theorem 3.1 is supposed to justify a unique orientation via a fourth point, but the statement is informal and the assumptions are not cleanly specified. Since the whole sequence factorization depends on a unique canonical order, this is not a cosmetic issue. If the theorem is central, it needs a cleaner statement in the main text. If it is not central, the paper should stop leaning on it rhetorically.

2. Theorem 3.1 is supposed to justify a unique orientation via a fourth point, but the statement is informal and the assumptions are not cleanly specified on Page 3. Since the whole sequence factorization depends on a unique canonical order, this is not a cosmetic issue. If the theorem is central, it needs a cleaner statement in the main text. If it is not central, the paper should stop leaning on it rhetorically.

3.  The introduction on Page 1 contrasts autoregressive models with diffusion models partly on computational efficiency and flexible generation. But the coordinate head in Section 3.3 still uses iterative denoising via diffusion loss. In other words, the method is not simply “AR instead of diffusion”; it is AR plus a diffusion-style coordinate generator at every step. Without runtime measurements, the efficiency claim is more slogan than evidence.

---

> ### Author Rebuttal · Authors · 2026-03-31
>
> Thank you for the careful reading. As requested, we added multi-seed statistics for the main results and matched-scale runtime comparisons, since these were the two points you identified as most relevant to your confidence in the paper.
> (Below, W = Weakness and Q = Question.)
>
> ***Q1. Multi-seed Stability***
>
> We now report multi-seed mean ± std for the main results in Tables 1, 2, and 3, and the conclusions remain stable across runs. For QM9 experiments, we ran 5 random seeds; for the other experiments, due to time and resource constraints, we ran 3 random seeds.
>
> | Dataset | Valid | Valid&Uni | AtomSta | MolSta |
> | --- | ---: | ---: | ---: | ---: |
> | QM9 | 97.37±0.14 | 92.29±0.20 | 99.31±0.05 | 94.51±0.26 |
> | GEOM-Drugs | 96.74±0.27 | N/A* | 86.97±0.26 | N/A* |
> | B3LYP | 99.07±0.21 | 98.55±0.25 | 84.69±0.12 | 23.96±0.40 |
>
> *For GEOM-Drugs, following the standard evaluation protocol used in Table 1 and prior work, we report only the shared metrics; Valid&Uni and MolSta are not part of this benchmark setup.
>
> For class-conditional generation, the five target classes also remain stable across seeds on all metrics:
>
> | Class | Hit Rate | Valid | Valid&Uni | AtomSta | MolSta |
> |-------|-----------|--------|------|----------|---------|
> | 7 | 90.34±0.62 | 98.76±0.31 | 92.18±0.54 | 99.62±0.14 | 96.89±0.48 |
> | 28 | 89.15±0.58 | 99.72±0.23 | 90.03±0.61 | 99.85±0.11 | 98.74±0.37 |
> | 3 | 85.08±0.71 | 99.81±0.17 | 86.24±0.58 | 99.87±0.09 | 98.96±0.42 |
> | 16 | 67.83±0.84 | 91.62±0.53 | 78.55±0.67 | 96.84±0.26 | 80.31±0.59 |
> | 23 | 81.12±0.65 | 99.48±0.28 | 82.03±0.56 | 99.86±0.12 | 98.67±0.44 |
>
> All metrics remain stable across seeds with small standard deviations. The gains are therefore not driven by a favorable single run.
>
> ***Q2. Runtime and Sampling Cost***
>
> We provide direct runtime measurements at matched parameter scale (~5.3M parameters):
>
> | Model | #Params | Sampling speed (samples/s) | Projected wall-clock for 10k QM9 generations (s) |
> | --- | ---: | ---: | ---: |
> | InertialAR | 5,346,393 | 37.70 | 265.22 |
> | Geo2Seq | 5,372,640 | 45.20 | 221.24 |
> | EDM | 5,340,921 | 1.70 | 5880.20 |
>
> InertialAR achieves 22.2× sampling speedup over EDM (37.70 vs. 1.70 samples/s), with projected wall-clock time of 265s vs. 5880s for 10K QM9 generations. Compared to the pure autoregressive baseline Geo2Seq, InertialAR is only ~20% slower (37.70 vs. 45.20 samples/s), demonstrating that the coordinate prediction module adds minimal overhead while delivering substantially better generation quality (Valid/MolSta in Tables 1–3). Training is also more efficient than EDM: InertialAR requires 125s per epoch versus 224.4s for EDM under the same batch size and GPU (1.8× speedup). These results directly support the efficiency motivation in the paper.
>
> ***W1 & W2. Theorem 3.1 and Unique Orientation***
>
> Theorem 3.1 in the main text is intended as a compressed paraphrase of the formal result. The formal statement with precise assumptions is given as Theorem D.1 in Appendix D, which provides the proof by contradiction showing why a fourth point with nonzero x- and y-components resolves the residual sign ambiguity of the inertial frame. In the revision, we will add an explicit forward reference to Appendix D and make these assumptions explicit in the main text. More broadly, we do not claim that inertial-frame canonicalization is theoretically perfect: Section 3.1 and Appendix E.1 already discuss the known edge cases, including degenerate eigenvalues and perturbation sensitivity, and show that these cases are negligible in practice on QM9 and GEOM-Drugs.
>
> ***W3. Efficiency Claim Without Runtime Evidence***
>
> The efficiency advantage stems from our architectural design. Our coordinate prediction module uses Diffusion Loss, but it is *not* a standard global diffusion model. The diffusion-style component in Section 3.3 is a lightweight per-token diffusion head: it predicts only the current atom's 3D coordinates conditioned on the autoregressive context, using a simple denoising MLP rather than repeated full-sequence attention over all atoms. In EDM, every denoising step reruns a global model over the entire molecule; in InertialAR, the Transformer captures token interdependence once through the sequence context, and the diffusion head models only the local conditional distribution of the current token. This architectural difference—avoiding repeated global denoising passes while retaining a diffusion-style objective for coordinate prediction—is what enables the 22.2× inference speedup shown in Q2. We will tighten the wording in revision so the efficiency claim is framed precisely as an advantage over full-molecule diffusion models, rather than as zero-overhead relative to every autoregressive alternative.
>
>
> We have provided the multi-seed stability and runtime evidence you requested, and we hope these additions strengthen confidence in the paper.

---

> > ### Author Rebuttal · Reviewer_Fgt2 · 2026-04-03
> >
> > Thank you for the detailed response and for providing further clarifications on the points raised. I appreciate the efforts made to address my initial concerns, and several aspects of the work are now clearer.
> >
> > However, I find the response regarding Theorem 3.1 to be unsatisfactory. My primary concerns is that,
> >
> > the theoretical claims in Theorem 3.1 currently lack the necessary rigor. When presenting formal theorems, it is essential that the assumptions and dependencies relied upon are clearly defined and mathematically sound.
> >
> > If the theorem serves as a cornerstone of the proposed method, it requires a more robust and precise treatment. Conversely, if the theorem is not central to the paper's primary contribution or its empirical success, the authors should consider whether its inclusion is necessary.

---

> > > ### Author Response · Authors · 2026-04-03
> > >
> > > Thank you for the follow-up. We agree that, as currently written, Theorem 3.1 in the main text is too compressed to serve as a formal theorem, because its assumptions and scope are not stated explicitly.
> > >
> > > To clarify, Theorem 3.1 is not the theoretical cornerstone on which the empirical results depend. Its intended role is much narrower: once the principal-axis frame is fixed up to sign, a fourth point with nonzero x- and y-components provides a sufficient condition for resolving the remaining sign ambiguity. The theorem concerns only this orientation-disambiguation step, not the full uniqueness of the canonical sequence.
> > >
> > > In the revision, we will therefore de-emphasize Theorem 3.1 in the main text and no longer present it as a formal cornerstone result. Concretely, we will replace the current main-text theorem with a brief geometric remark, and rewrite the appendix presentation accordingly with explicit assumptions, including distinct principal moments, deterministic anchor selection, and the nonzero x/y anchor condition in the principal-axis basis. We will also revise the surrounding text so that the paper does not rhetorically lean on this argument as a statement of globally unique canonicalization.
> > >
> > > Concretely, the revised appendix statement will be along the following lines: let Q be a right-handed principal-axis basis with fixed axis ordering, and assume the inertia tensor has three distinct eigenvalues ($\lambda_1 < \lambda_2 < \lambda_3$). Among the four sign assignments of Q that preserve right-handedness, if a deterministically selected anchor point v has nonzero x- and y-coordinates in the Q-basis, then there exists exactly one sign assignment under which v lies in the first quadrant (x > 0, y > 0). This yields a unique resolution of the residual sign ambiguity of the principal-axis frame under the stated assumptions.
> > >
> > > We hope this clarification resolves your remaining concern about the role and rigor of Theorem 3.1, and makes the paper easier to assess based on its actual methodological and empirical contributions.

---

### Decision · Program_Chairs · 2026-04-30

**Decision:**

Accept (regular)

**Comment:**

This is a borderline paper, with all reviewers rating "weak accept". All reviewers agreed that the work is technically sound, with mixed (though positive) views on its originality and significance. The paper introduces a system, InertialAR, which incorporates several novel innovations, which may be meaningfully used or built on in future work. Ablation studies in the appendix are clear.

There are in my view two important lingering concerns:

* The canonicalisation is not necessarily unique, in edge cases (though empirically this seems fine)
* The evaluation, particularly in the original submission, focuses on validity and stability metrics; these are somewhat limited in their scope, and note that most methods perform well here on e.g. QM9 (standard errors on results tables 1 through 3 would also be appreciated…). However, the hit rate metrics for conditional generation are appreciated by reviewers (and myself); as are the synthetic accessibility metrics provided in the rebuttal (with the caveat that this score is not truly representative of synthesizability)

Overall, on balance the authors have responded quite in detail to the reviewer concerns, and I would agree with the "weak accept" judgement for the paper in its originally submitted form.

For a camera-ready, please incorporate the discussion and additional results from the rebuttal into the final draft.